# Self-supervised Masked Graph Autoencoder via Structure-aware Curriculum

## Abstract

Self-supervised learning (SSL) on graph-structured data has attracted considerable attention recently. Masked graph autoencoder, as one promising generative graph SSL approach that aims to recover masked parts of the input graph data, has shown great success on various downstream graph tasks. However, existing masked graph autoencoders fail to consider the degrees of difficulties of recovering the masked edges that often have different impacts on the model performance, resulting in suboptimal node representations. To tackle this challenge, in this paper, we propose a novel curriculum based self-supervised masked graph autoencoder that is able to capture and leverage the underlying degree of difficulties of data dependencies hidden in edges, and design better mask-reconstruction pretext tasks for learning informative node representations. Specifically, we first design a difficulty measurer to identify the underlying structural degree of difficulties of edges during the masking step. Then, we adopt a self-paced scheduler to determine the order of masking edges, which encourages the graph encoder to learn from easy parts to difficult parts. Finally, the masked edges are gradually incorporated into the reconstruction pretext task, leading to high-quality node representations. Experiments on several real-world node classification and link prediction datasets demonstrate the superiority of our proposed method over state-of-the-art graph self-supervised learning baselines. This work is the first study of curriculum strategy for masked graph autoencoders, to the best of our knowledge.

## 1 Introduction

Graph structured data is ubiquitous in the real world, such as social networks, citation networks, e-commerce networks, etc. Recently, graph neural networks (GNNs) have gained considerable attention for learning representations pertinent to graph-structured data and have shown great success in supervised learning (Xu et al., 2019) or semi-supervised learning (Kipf & Welling, 2017; Hamilton et al., 2017), where task-specific labels are necessary to train GNNs as the supervision information. However, collecting numerous task-specific annotations is a challenging endeavor in practice.

Self-supervised learning (SSL), as one type of unsupervised methods that are popular in computer vision and natural language processing (Chen et al., 2020; He et al., 2020), has also revolutionized the field of graph learning due to its promising performances in learning informative representations through well-designed pretext tasks without relying on labels. Current graph SSL approaches can be broadly categorized into two classes, i.e., contrastive and generative approaches. Specifically, contrastive graph SSL has firstly emerged as the prevailing method, adopting the instance discrimination as the pretext task, e.g., DGI (Veličković et al., 2019), MVGRL (Hassani & Khasahmadi, 2020), and BGRL (Thakoor et al., 2022), etc. However, although there are some inspiring recent works exploring augmentation-free strategies like AFGRL (Lee et al., 2022b) and IGCL (Li et al., 2023a), most classical contrastive graph SSL methods heavily rely on well-designed heuristic graph augmentation, so that their performances can substantially degenerate when the graph augmentation is not compatible with downstream tasks (Zhang et al., 2021b). Generative graph SSL approaches can naturally tackle this problem by adopting one simple yet effective strategy as the pretext task, which is to directly reconstruct the missing parts of the input graph data, whose representative methods include GPT-GNN (Hu et al., 2020b), GraphMAE (Hou et al., 2022), S2GAE (Tan et al., 2023). They achieve promising performances without relying on the high-quality hand-crafted graph augmentation.

Despite the noticeable success, the existing generative graph SSL literature fails to consider the influence of the difficulties of pretext tasks during training process and merely treats all data samples for training equally, which can lead to suboptimal performances. Intuitively, the pretext tasks should be designed from dealing with easy data samples first and gradually to harder samples. This is because tackling much more difficult data samples at an early training stage will inevitably make the initial GNN encoder, with randomly initialized parameters, struggle to finish the pretext tasks. Similarly, feeding over easy data samples at the later training stage will also bring limited benefits to the GNN encoder. However, it remains unexplored to design tailored easy-to-hard pretext tasks for more powerful representations, which poses the following challenges.

- It is technically difficult to design tailored reconstruction tasks to encourage the GNNs to capture informative patterns of the input graph into representations.

- It is challenging to derive a proper principle to quantify the difficulty of reconstruction samples for training the GNNs.

- It is also non-trivial to design a feasible schedule strategy to gradually exploiting data samples for reconstruction by explicitly considering the training status of GNNs.

To tackle these challenges, we propose the Curriculum Masked Graph AutoEncoder (**Cur-MGAE**[1]) to capture and leverage the underlying degree of difficulties of data dependencies hidden in edges and design better mask-reconstruction pretext tasks for learning informative representations. The proposed method is capable of learning GNNs for informative representations in a tailored easy-to-hard meaningful order to incorporate training data samples for reconstruction pretext tasks. Specifically, we first propose a tailored structure-aware edge reconstruction task, namely recovering purposely masked edge from the other unmasked graph structure. By the pretext tasks, the GNN encoder can extract useful patterns from the input graph into the output representations. Then, we advocate for a self-supervised method to select the top $\mathcal{K}$ easiest edges that the GNN encoder anticipates to reconstruct, so that the difficulties of reconstruction samples can be quantified effectively. Furthermore, we propose a tailored self-paced learning strategy to gradually exploit the edges for reconstruction as the training processes. The task of recovering the missing structures for informative node representations and selecting appropriate edges for training are integrated into a joint training framework. Finally, the GNN encoder can be trained by feeding data samples in a meaningful order to select proper data samples in pretext tasks, whose difficulties are well aligned with the GNN's training status, leading to more powerful node representations and showing better performances in downstream tasks.

We theoretically analyze the convergence guarantee of this tailored training paradigm by proving the properties of the avoidance of saddle points and the second-order convergence. Extensive experiments on various real-world node classification and link prediction benchmarks demonstrate that our proposed model can achieve significant performance gains against state-of-the-art approaches.

The main contributions of this paper are summarized as follows:

- We propose a novel method to train the GNN encoder by feeding the data with a tailored easy-to-hard meaningful order to design pretext tasks. To the best of our knowledge, this is the first attempt to study curriculum graph self-supervised learning.

- We propose a joint framework to recover the missing edges of the input based on the unmasked graph structure and schedule the training edges for reconstruction in a proposed self-paced learning manner, so that the GNN encoder can be trained more effectively.

- Theoretical analysis of the convergence properties of the proposed **Cur-MGAE** model and extensive experiments show that our proposed method significantly outperforms the state-of-the-art graph SSL approaches, covering both contrastive and generative approaches.

## 2 RELATED WORK

**Graph Neural Network.** Graph-structured datasets are prevalent across real-world scenarios (Hu et al., 2020a; Li et al., 2019a; 2021). Recently, the advent of graph neural networks (GNNs) (Kipf & Welling, 2017; Veličković et al., 2018; Xu et al., 2019) has ushered in transformative changes to the

---

[1]We will release source code at publication time.

realm of graph representation learning (Zhang et al., 2020). Exhibiting robust outcomes across diverse tasks such as node categorization (Kipf & Welling, 2017), link inference (Zhang & Chen, 2018), and whole-graph classification (Xu et al., 2019), GNNs have achieved noteworthy accomplishments in demanding domains, encompassing areas like pharmaceutical research (Wu et al., 2018), protein activity forecasting (Jiang et al., 2017), and traffic prediction (Jiang & Luo, 2021). Typically, GNNs employ a paradigm rooted in neighborhood aggregation or message passing, where the representation of a node is progressively refined by assimilating representations from neighboring nodes (Veličković et al., 2018; Xu et al., 2019). Nonetheless, to attain top-tier results, several renowned GNN models (Ji et al., 2021; Fan et al., 2021; Gao et al., 2021; Miao et al., 2021; Ye & Ji, 2021; Zhang et al., 2021c) necessitate end-to-end training utilizing task-specific annotations, which may be in short supply for certain graph datasets. In contrast to these label-reliant models, our introduced model leverages self-supervised generative techniques, significantly diminishing the reliance on manually curated labels which are the vital factors in graph representation learning.

**Graph Self-Supervised Learning.** Graph self-supervised learning (SSL) techniques (Liu et al., 2022; You et al., 2020; Peng et al., 2020; Xu et al., 2021; Sun et al., 2023b) primarily branch into contrastive and generative methodologies. Recently, contrastive graph SSL first becomes popular. Some methods mainly explore negative sampling strategies like corruption-based negative pair construction in DGI (Veličković et al., 2019), and in-batch negatives in methods like GCA (Zhu et al., 2021). In these methods, graph augmentation is vital for the efficacy of contrastive learning. Nevertheless, the understanding of augmentation in graph contexts lacks clarity and theoretical grounding, making their label-invariance and optimality questionable. On the other hand, generative SSL focuses on replenishing omitted portions of input data, bifurcating into autoregressive and autoencoding models. While historically generative approaches lag in performance compared to their contrastive counterparts, there are notable works in autoregressive graph models, such as GPT-GNN (Hu et al., 2020b). In autoencoder space, models like GAE and VGAE (Kipf & Welling, 2016) set early benchmarks. More recent methods are developed such as GraphMAE (Hou et al., 2022), GraphMAE2 (Hou et al., 2023), GigaMAE (Shi et al., 2023), SeeGera (Li et al., 2023e), RARE (Tu et al., 2023), S2GAE (Tan et al., 2023) and Bandana (Zhao et al., 2024), whose key idea is to reconstruct the masked parts based on the left unmasked graph structures. However, these methods largely ignore the influence of the difficulties of self-supervised tasks and simply treat all training samples equally, which can lead to suboptimal performances on downstream tasks. One of the recent methods, AUG-MAE(Wang et al., 2024) designs an easy-to-hard adversarial masking strategy on nodes to provide hard-to-align samples, which improves the alignment performance.

**Curriculum Learning.** Curriculum Learning (CL) is an approach where training progresses from simpler to more complex tasks, mimicking the manner in which humans often approach learning through courses taught in school (Bengio et al., 2009; Wang et al., 2021a). The foundational algorithm in this domain is termed "Baby Step" (Spitkovsky et al., 2010), responsible for dictating the complexity and sequential order of data inputs. Subsequent advancements led to the inception of the self-paced method (Kumar et al., 2010), designed to autonomously curate data samples based on training loss metrics. In addition, numerous automatic CL frameworks have emerged, such as transfer teacher (Hacohen & Weinshall, 2019), reinforcement learning teacher (Zhao et al., 2020), as well as others tailored to specific datasets, models, and objectives (Sinha et al., 2020). Notably, there's also been an integration of CL with areas like disentangled recommendations (Chen et al., 2021), combinatorial optimization (Zhang et al., 2022b), neural architecture search (Zhou et al., 2022), and video grounding (Lan et al., 2023). Some works also introduce CL into graphs (Li et al., 2023c), e.g., GNN-CL (Li et al., 2024), CurGraph (Wang et al., 2021b), etc. Central to the CL paradigm are mechanisms that evaluate the intricacy of data samples and orchestrate training regimens, determining either the sequence or the relative importance of data segments. However, these methods generally need labels as the supervised information for training encoder and ignore the self-supervised scenarios where the labels are not available. More importantly, the existing methods consider the training samples as independent samples, while we focus on more challenging structure-aware curriculum learning where samples are inter-connected with each other so that they can not be treated as independent samples.

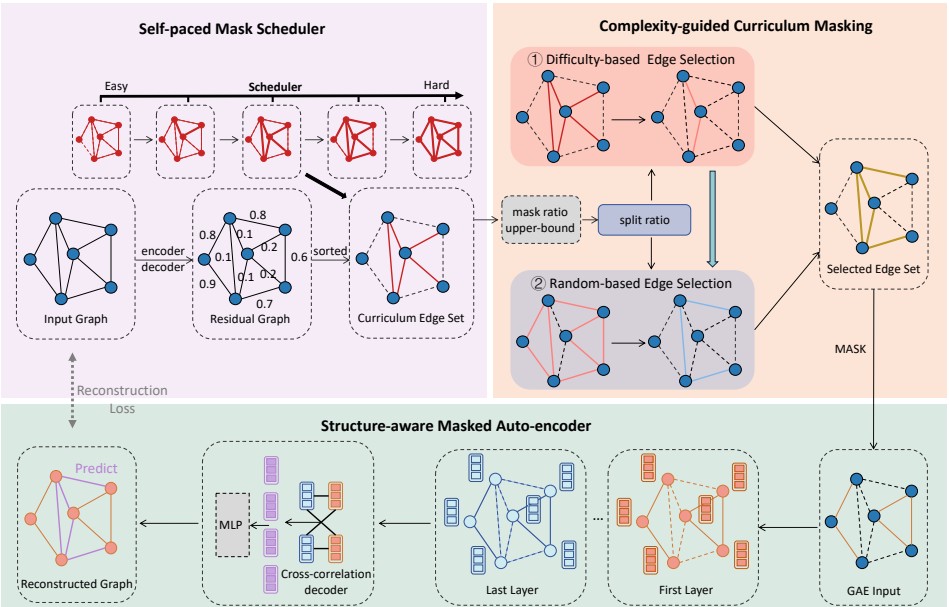

Figure 1: The framework of our proposed **Cur-MGAE** method. Given one input graph, we first propose a complexity-guided curriculum masking to select the edges to be masked, where each edge has its reconstruction residual error representing the difficulty score. Then, we propose a self-paced mask scheduler to determine the curriculum based on the training stage. Finally, we propose a structure-aware masked auto-encoder to reconstruct the structures for self-supervised learning.

## 3 METHOD

In this section, we introduce our proposed **Cur-MGAE** model, which is able to learn informative representations with a structure-aware curriculum in a self-supervised fashion. Specifically, we propose a structure-aware masked auto-encoder, a complexity-guided curriculum masking, and a self-paced mask scheduler. The framework of our method is shown in Figure 1.

### 3.1 STRUCTURE-AWARE MASKED AUTO-ENCODER

In this section, by designing an edge reconstruction task, we propose a structure-aware masked auto-encoder to learn informative node representations without supervised labels.

**GNN encoder.** Denote a graph as $\mathcal{G}(\mathcal{V}, \mathcal{E})$, where $\mathcal{V}$ and $\mathcal{E}$ denote the set of nodes and edges respectively. It can also be expressed as $\mathcal{G} = (\mathbf{H}, \mathbf{A})$ with the node representations $\mathbf{H}$ and adjacency matrix $\mathbf{A}$. We adopt a GNN to encode the graph and obtain the node representations, i.e.,

$$h_v^{(k)} = \text{COM}(h_v^{(k-1)}, \text{AGG}(h_u^{(k-1)} : u \in N_v)), \tag{1}$$

where $h_v^{(k)}$ is the embedding of node $v$ at $k$-th layer, and $N_v = \{u : (v, u) \in \mathcal{E}\}$ is the set of direct neighbors of node $v$, $\text{AGG}(\cdot)$ is the message aggregation function to aggregate the neighborhood information, and $\text{COM}(\cdot)$ is the combination function to update the node embedding with the aggregated messages. Multiple GNN layers are stacked so that for any node $v$, we can obtain $K$ node representations $h_v^{(1)}, h_v^{(2)}, ..., h_v^{(K)}$, where each embedding captures the neighbor structure within $K$-hop neighborhood. We denote the node representations encoded by the GNN as: $\mathbf{H} \leftarrow \text{ENC}(\mathbf{H}, \mathbf{A}) \in \mathbb{R}^{N \times d}$, where $N$ is the number of nodes, $d$ is dimensionality, $\text{ENC}(\cdot)$ is the encoder.

**Cross-correlation decoder.** After obtaining the node embeddings by the GNN encoder, we design a cross-correlation decoder to fully consider the inherent similarity between nodes to reconstruct the structures. We first obtain the edge embeddings by:

$$h_{e_{v,u}} = ||_{k,j=1}^{K} h_v^{(k)} \odot h_u^{(j)}, \tag{2}$$

where $\odot$ represent the element-wise product, $h_{e_{v,u}} \in R^{dK^2}$ is the final edge representation, $K$ is the number of layers, $||$ is the concatenation operator. In this way, the shared attributes between adjacent elements are accentuated, while the divergent components are eradicated via element-wise multiplication. Consequently, only those elements that exhibit a high degree of correlation between neighboring nodes are retained in the representation $h_{e_{v,u}}$. The amplification of shared aspects within each edge representation is instrumental in facilitating edge prediction. Following the generation of such an embedding, we employ a Multilayer Perceptron (MLP) layer with a sigmoid activation function to predict the probability of existence for each edge, formally expressed as $g(v,u) = MLP(h_{e_{v,u}})$. The generation of edge representation plays a pivotal role in filtering out noisy information, thereby only common features are preserved. This selective retention of information is beneficial in guiding the MLP layer towards making accurate predictions. This approach effectively streamlines the decision-making process by focusing on the most relevant and shared characteristics, thereby enhancing the overall efficiency and accuracy of edge prediction. We denote the structures obtained by the decoder as: $\mathbf{A} \leftarrow \text{DEC}(\mathbf{H}) \in \mathbb{R}^{N \times N}$.

**Reconstruction task.** We adopt a perturbation-then-reconstruct paradigm for self-supervised learning, to enhance the quality of the learned node representations. Specifically, we select a part of the edges to be masked in the original graph to get the perturbed graph: $\tilde{\mathbf{A}} = \mathbf{A} - \mathbf{A}_{\text{mask}}$. We denote $\mathbf{A}_{\text{mask}}$ as the adjacency matrix of the masked edges $\mathcal{E}_{mask}$. Then we adopt the reconstruction loss as the supervised signal: $\mathcal{L}_{SSL} = \ell(\mathbf{A}, \text{DEC}(\text{ENC}(\mathbf{H}, \tilde{\mathbf{A}})))$, where $\ell(\cdot)$ is the loss function. We implement it as the cross-entropy function, and obtain the reconstruction objective as:

$$\mathcal{L}_{SSL} = -\frac{1}{|\mathcal{E}_{mask}|} \sum_{(v,u) \in \mathcal{E}_{mask}} log \frac{exp(g(v,u))}{\sum_{z \in V} exp(g(v,z))}. \tag{3}$$

$g(\cdot)$ is the predicted probability of the presence of an existing edge, namely $g(v,u) = MLP(||_{k,j=1}^{K} h_v^{(k)} \odot h_u^{(j)})$, wherein $h_u^{(j)}$ and $h_v^{(k)}$ denote the $j^{th}/k^{th}$ hidden representation of node $u$ and $v$, respectively. In this context, the learned representations are deemed to be rich in information since they are necessitated to encapsulate ample structural and attribute information that is sufficient to reconstruct the original structures from the perturbed graphs. The structures can be deeply encoded into the node representations, thereby providing a robust representation for the subsequent adoption in downstream tasks.

## 3.2 Complexity-guided Curriculum Masking

Since the roles of different edges in a graph could be different, randomly sampling the mask edges to construct a perturbed graph may introduce optimization difficulties during the reconstruction process. For example, a perturbed graph with little structural information may be too hard for the GNN to reconstruct the original structures at a early training stage, where the encoded node embedding may not be well-trained. To address the issue, we propose a complexity-guided curriculum masking module to select edges from easy to hard, so that the optimization process can be smoothed.

Specifically, we try to identify which edges are explicitly important to the graph and gradually filter them out to increase the learning difficulty. We formally defined the edge difficulty as: the difficulty score measuring how hard for the current model to predict the edge correctly. To quantify the difficulty of each edge in a given graph, we first use the current model to reconstruct the original graph as: $\mathbf{A}_{re} = \text{DEC}(\text{ENC}(\mathbf{H}, \mathbf{A}))$. The predicted structures $\mathbf{A}_{re}$ reflect the edges that are expected to be inherently drawn from the current model, where the edge probabilities can be viewed as the model's confidence for the edge occurrence. Intuitively, lower edge confidence indicates that the edge is more complex to be reconstructed by the current model. So we propose to leverage structural complexity to estimate the hardness of the edge for curriculum learning. Specifically, we calculate the residual errors as: $\mathbf{R} = \mathbf{A} - \mathbf{A}_{re}$. The residual errors could also potentially serve as a representation of the degree of confidence the model possesses regarding the existence of an edge between two nodes. It measures the distance between the predicted structure and the original structure, which tells how difficult the edges are for the current model. Consequently, a smaller residual error could be interpreted as an indication of a more pronounced significance of the edge existence. This implies that such edges are relatively easier to learn and comprehend within the context of the model. Moreover, by implementing a masking technique on the smallest $\mathcal{K}$ edges, the complexity of the task for the graph learning model can be substantially reduced. This is due to the fact that the model will be

primarily focused on learning the fundamental characteristics and properties of these $\mathcal{K}$ edges, which are relatively easier to learn due to their lower residual errors. Therefore, this strategy could enhance the overall efficiency and effectiveness of the learning process within our model.

### 3.3 SELF-PACED MASK SCHEDULER

In this section, we propose a self-paced mask scheduler to progressively and autonomously incorporate an increasing number of edges throughout the training process. One potential solution could be incrementally increasing the value of $\mathcal{K}$ during the training process. However, identifying and updating an appropriate $\mathcal{K}$ value within the process is non-trivial. Moreover, the selection of edges directly correlates to a discrete optimization problem within an exceedingly vast topological space (1 for selecting and 0 for not selecting). This factor significantly complicates the optimization process. To address this issue, we first relax each item in $\mathbf{S}^{(\mathbf{t})}$ into $[0,1]$, and transform the problem into a continuous constrained optimization problem. Specifically, we regard the constraint as a Lagrange multiplier, and introduce a regularization term into the loss function as follows: $f(\mathbf{S}; \lambda, \mathbf{A}) = \lambda ||\mathbf{S}^{(\mathbf{t})} \odot \mathbf{A} - \mathbf{A}||$. $\mathbf{S}^{(\mathbf{t})}$ is the edge selection matrix at training iteration $t$ in the self-paced scheduler, which has the same dimensionalities as the adjacency matrix $\mathbf{A}$. To optimize the loss function, the edge selection matrix $\mathbf{S}^{(\mathbf{t})}$ is relaxed to be continuous. Each element in the optimized $\mathbf{S}^{(\mathbf{t})}$ will belongs to $[0,1]$, and then, a cutoff threshold of 0.5 will be applied for converting the continuous value in edge selection matrix $\mathbf{S}^{(\mathbf{t})}$ back into discrete binary value, or formally written as $\{0,1\}$, so that the selected edges for training at step $t$ is $\mathbf{A}^{(\mathbf{t})} = \mathbf{S}^{(\mathbf{t})} \odot \mathbf{A}$, where $\odot$ means the element-wise multiplication. The regularization term in the process of optimization, endeavors to mask as many edges as possible, contingent upon the parameter $\lambda$. By gradually increasing the coefficient $\lambda$, more edges are forced to be selected until the selected training edges $\mathbf{A}^{(\mathbf{t})}$ converges to the original adjacency matrix $\mathbf{A}$. Therefore, it can be regarded as scheduling the training edges in an easy-to-hard manner. After adding this term, the loss function for our self-paced mask scheduler is:

$$\mathcal{L}_{SPCL} = \beta \sum_{i,j} S_{ij} R_{ij} + f(\mathbf{S}; \lambda, \mathbf{A}), \tag{4}$$

where $\beta$ is the balancing hyper-parameter, and $\mathbf{S_{ij}}$ is used for extracting masked edges. $R_{ij} = ||\mathbf{A_{ij}} - \tilde{\mathbf{A}}_{\mathbf{ij}}^{(\mathbf{t})}||$ represent the edge residual, $\mathbf{A_{ij}}$ denotes the training structure, $\tilde{\mathbf{A}}_{\mathbf{ij}}^{(\mathbf{t})}$ represents the predicted structure and $|| \cdot ||$ is chosen as the squared $l_2$ norm.

However, entirely rely on the assessed difficulty level for edge selection will invariably masks easy edges during the training phase, potentially compromising the model's generalizability due to entrapment within such edges. To mitigate this, *split ratio* is introduced to facilitate random edge selection. A smaller value for this parameter implies more edges are chosen randomly. Essentially, this hyperparameter aims to strike a balance between exploration and exploitation. Consequently, overfitting can be solved through adjustment of such hyperparameter.

### 3.4 OVERALL PIPELINE

As mentioned in above sections, our proposed model necessitates the minimization of $\mathcal{L}_{all}$, which comprises two sets of parameters for optimization, and lead to a difficult bi-level optimization problem. To address this issue, we propose an optimization algorithm, which sequentially trains two individual self-supervised models with the corresponding objectives. The overall loss function can be formulated as follows:

$$\mathcal{L}_{all} = \mathcal{L}_{SSL} + \mathcal{L}_{SPCL}, \tag{5}$$

During the training process, to smooth the transition between each training iteration, regularizers $\frac{\gamma}{2}||\boldsymbol{w} - \boldsymbol{w}^{(t-1)}||$, and $\frac{\gamma}{2}||\boldsymbol{S} - \boldsymbol{S}^{(t-1)}||$ are added in each optimization process. The overall pipeline of our proposed method is summarized in Algorithm 1.

### 3.5 THEORETICAL ANALYSES

We provide the theoretical analyses on the convergence of our method in Theorem 1 and 2. Due to the page limit, the prove of the theorems is given in Appendix B.

---

**Algorithm 1:** The optimization process of **Cur-MGAE**

---

**Data:** Node features X, adjacency matrix $A$, step size $\mu$, and hyperparameter $\gamma$
**Result:** Trained GNN model parameter $w$
Initialize $w^{(0)}$, $S^{(0)}$, and $\lambda^{(0)}$;
Compute $A^{(0)} = S^{(0)} \odot A$;
**while** *Not Converged* **do**

    $w^{(t)} = argmin_{w}\mathcal{L}_{SSL}(X, A^{(t-1)}; w) + \frac{\gamma}{2}||w - w^{(t-1)}||$;

    Encode training structure $A$ to embedding $Z^{(t)}$ based on updated GNN model $f$;

    For all pairs of $i$ and $j$, predict the existence of edge $A_{ij}$ as $\tilde{A}_{ij}^{(t)} = g(z_i^{(t)}, z_j^{(t)})$;

    Relax $S^{(t)}$ to be continuous and optimize it as $S^{(t)} = argmin_{S}\mathcal{L}_{SPCL} + \frac{\gamma}{2}||S - S^{(t-1)}||$;

    $\mathcal{K} = |S^{(t)} \geq 0.5|$;
    **if** $\mathcal{K} \geq mask\ ratio * |\mathcal{E}|$ **then**
        | $\mathcal{K} = mask\ ratio * |\mathcal{E}|$;
    **else**
        | Update $\lambda$ based on designed curriculum pace;

    Choose $split\ ratio * \mathcal{K}$ edges with largest $S_{ij}$ to be $S_d^t$;
    Choose the remaining edges randomly and denote as $S_r^t$;
    $S^{(t)} = S_d^t + S_r^t$;
    Update $A^{(t)} = S^{(t)} \odot A$;

---

**Theorem 1 (Avoidance of Saddle Points)** *For a sufficiently large $\gamma$, if the second derivatives of $\mathcal{L}_{SSL}(X, A^{(t-1)}; w)$ and $f(S; \lambda)$ are continuous, any bounded sequence $(w^{(t)}, S^{(t)})$ generated by Algorithm 1 with random initialization will not converge to a strict saddle point of F almost surely.*

**Theorem 2 (Second Order Convergence)** *For a sufficiently large $\gamma$, if the second derivatives of $\mathcal{L}_{SSL}(X, A^{(t-1)}; w)$ and $f(S; \lambda)$ are continuous and $\mathcal{L}_{SSL}(X, A^{(t-1)}; w)$ and $f(S; \lambda)$ satisfy the Kuradyka-Lojasiewicz (KL) property (Wang et al., 2022b), any bounded sequence $(w^{(t)}, S^{(t)})$ generated by Algorithm 1 with random initialization will almost surely converge to a second-order stationary point of F.*

## 4 EXPERIMENT

In this section, we conduct experiments to verify the effectiveness of our proposed **Cur-MGAE** method, including experimental setup, quantitative comparisons on both node classification and link prediction datasets, and more deep analyses. More experiments are included in Appedix H.

### 4.1 EXPERIMENTAL SETUP

**Datasets.** For node classification, we consider 3 Planetoid datasets (Cora, Citeseer and Pubmed (Sen et al., 2008)), and 3 common datasets including Coauthor-CS (Shchur et al., 2019), Coauthor-Physics (Shchur et al., 2019), and OGBN-arxiv (Hu et al., 2020a). We evaluate the model performance based on accuracy (%) scores. For link prediction, we consider the 3 Planetoid benchmarks (Cora, Citeseer and Pubmed (Sen et al., 2008)), as well as datasets from more challenging OGB benchmarks (Hu et al., 2021), i.e., OGBN-ddi, OGBL-collab, and OGBL-ppa. We report the AUC (%) score (Bradley, 1997) for the small-scale Cora and Citeseer, and Hit rate (Hits@N) for OGB datasets, following the work (Tan et al., 2023) for fair comparisons.

**Baselines.** We compare the proposed **Cur-MGAE** with the following two groups of baselines. One group of baselines are state-of-the-art contrastive graph SSL methods including DGI (Veličković et al., 2019), GIC (Mavromatis & Karypis, 2021), MVGRL (Hassani & Khasahmadi, 2020), and BGRL (Thakoor et al., 2022). The other group of baselines are generative graph SSL methods including GAE (Kipf & Welling, 2016), GraphSAGE (Hamilton et al., 2017), ARGVA (Pan et al., 2019), GPT-GNN (Hu et al., 2020b), RRL (Zhu et al., 2020), GraphMAE (Hou et al., 2022), GraphMAE2 (Hou et al., 2023), MaskGAE (Li et al., 2023d), Bandana (Zhao et al., 2024), AUG-MAE (Wang et al., 2024), and S2GAE (Tan et al., 2023).

## 4.2 EXPERIMENTAL RESULTS

**Node Classification Results.** Table 1 summarizes the results of our method and baselines. Our proposed **Cur-MGAE** outperforms existing contrastive and generative baselines with a higher average rank, demonstrating the superiority of scheduling the training data, whose order is determined by the difficulty measurer in the reconstruction pretext tasks, can benefit encoding useful information of nodes into representations. For example, **Cur-MGAE** increases the classification accuracy by nearly 1% against the strongest baseline on the representative node classification benchmark OGBN-arxiv.

Table 1: Node classification accuracy (%) of our proposed method and baselines. In each column, the boldfaced score denotes the best result of all the methods. Rank is the average rank. Our method achieves the best performances in terms of average rank.

| Dataset | Cora | Citeseer | Pubmed | Coauthor-CS | Coauthor-Physics | OGBN-arxiv | Rank |
|---|---|---|---|---|---|---|---|
| DGI | 85.41 ± 0.34 | 74.51 ± 0.51 | 76.80 ± 0.60 | 92.77 ± 0.38 | 94.55 ± 0.13 | 67.08 ± 0.43 | 9.50 |
| GIC | **87.70 ± 0.01** | **76.39 ± 0.02** | 77.40 ± 1.9 | 91.33 ± 0.30 | 93.49 ± 0.42 | 64.00 ± 0.22 | 9.17 |
| MVGRL | 85.86 ± 0.15 | 73.18 ± 0.22 | 80.10 ± 0.70 | 92.87 ± 0.13 | 95.35 ± 0.08 | 68.33 ± 0.32 | 8.42 |
| BGRL | 86.16 ± 0.20 | 73.96 ± 0.14 | 82.05 ± 0.85 | 93.35 ± 0.06 | **96.16 ± 0.09** | 71.77 ± 0.19 | 4.00 |
| GCN | 83.60 ± 0.52 | 63.37 ± 1.21 | 78.23 ± 1.63 | 89.79 ± 0.09 | 93.26 ± 0.05 | 66.01 ± 0.37 | 13.67 |
| GraphSage | 74.30 ± 1.84 | 60.20 ± 2.15 | 81.96 ± 0.74 | 89.74 ± 0.19 | 93.35 ± 0.06 | 64.79 ± 2.91 | 13.00 |
| ARGVA | 85.86 ± 0.72 | 73.10 ± 0.86 | 81.51 ± 1.00 | 84.68 ± 0.26 | 92.89 ± 0.11 | 50.06 ± 1.21 | 12.08 |
| GPT-GNN | 84.69 ± 0.09 | 71.82 ± 0.13 | 81.45 ± 0.18 | 91.07 ± 0.21 | 95.02 ± 0.15 | 70.16 ± 0.10 | 10.33 |
| RRL | 57.29 ± 0.13 | 59.57 ± 1.77 | 75.06 ± 0.37 | 84.71 ± 0.95 | 94.90 ± 0.02 | 66.36 ± 0.13 | 14.33 |
| GraphMAE | 85.45 ± 0.40 | 72.48 ± 0.77 | 81.10 ± 0.40 | **93.47 ± 0.04** | 96.13 ± 0.03 | 71.86 ± 0.00 | 6.50 |
| GraphMAE2 | 84.50 ± 0.60 | 73.40 ± 0.30 | 81.40 ± 0.50 | 92.13 ± 0.12 | 95.44 ± 0.08 | 71.89 ± 0.03 | 8.25 |
| MaskGAE | 87.31 ± 0.05 | 75.20 ± 0.07 | 83.58 ± 0.45 | 92.31 ± 0.05 | 95.79 ± 0.02 | 70.99 ± 0.12 | 4.50 |
| Bandana | 84.62 ± 0.37 | 73.60 ± 0.16 | 83.53 ± 0.51 | 93.10 ± 0.05 | 95.57 ± 0.04 | 71.09 ± 0.24 | 6.33 |
| AUG-MAE | 84.30 ± 0.40 | 73.20 ± 0.40 | 81.40 ± 0.40 | 92.15 ± 0.22 | 95.34 ± 0.60 | 71.90 ± 0.20 | 8.58 |
| S2GAE | 86.15 ± 0.25 | 74.60 ± 0.06 | 84.19 ± 0.21 | 91.70 ± 0.08 | 95.82 ± 0.03 | 72.02 ± 0.05 | 4.50 |
| **Cur-MGAE** | 87.25 ± 0.55 | 74.68 ± 0.37 | **85.86 ± 0.14** | 92.69 ± 0.17 | 95.91 ± 0.05 | **73.00 ± 0.06** | 2.83 |

Table 2 shows the link prediction results of our method and baselines[2]. From the results, we can observe that generative graph SSL methods (e.g., GraphMAE, MaskGAE, S2GAE) generally outperform the contrastive graph SSL methods, which verify the effectiveness of the key design of the pretext task, i.e., recovering the masked parts based on the unmasked left parts. Our curriculum method **Cur-MGAE** achieves the best results in 2 out of 6 datasets, while reporting the comparable performances in the remaining 4 datasets. For example, **Cur-MGAE** increases the performances by nearly 3% against the strongest baselines on OGBL-collab and OGBL-ppa. We attribute the results to the fact that these baselines fail to consider the influence of the difficulties of pretext tasks during training process and merely treat all data samples for training equally, leading to suboptimal performances. One example is MaskGAE (Li et al., 2023d), which is one competitive method which seeks to reconstruct the masked edges and node degrees jointly. It performing well in some small-scale datasets but showing poor results in the large-scale datasets. One plausible reason is that MaskGAE ignores the degrees of difficulties of recovering the masked edges that often have different impacts on the model performance during training process. Therefore, it have unsatisfactory performances on more large-scale benchmarks, where the informative node representations are harder to be learned. In contrast, our method designs a tailored reconstruction pretext task by feeding the data samples via an easy-to-hard meaningful order into training process. None of the baselines is consistently competitive across all of the datasets, as opposed to our method.

## 4.3 VISUALIZATION OF LEARNED EDGE SELECTION CURRICULUM

To visually show the edge selection strategy, instead of using real datasets, we built synthetic datasets with ground truth edge difficulty labels following previous work (Karimi et al., 2018; Abu-El-Haija et al., 2019). Every conceived graph encompasses 5,000 nodes segmented into 10 node groups, nodes in each group are of the same amount and labeled from 1 to 10, respectively. A visualization of the synthetic dataset is provided in Appendix G. These node attributes are derived from intersecting multi-Gaussian distributions, indicating the position of each node, and nodes are classified into 10 labels based on the node feature. The edge difficulty is correlated with the node labels. Specifically, we believe the edges between two nodes with the same label are *easy* to identify, between two nodes with neighboring labels have *medium* difficulties to be learned, and between two faraway nodes as *hard* to predict. Each generated synthetic dataset is associated to a *homophily coefficient* (*homo*) to

---

[2]Note that GraphMAE2 and AUG-MAE are omitted here since they are node or graph classification methods and are not designed for link prediction tasks.

Table 2: Link prediction results (%) of our proposed method and baselines. Our method achieves strong performance gains compared with the baselines no matter on small-scale datasets or more challenging large-scale benchmarks. Note that "–" denotes out-of-memory (24G) in experiments, "/" denotes that the current model do not support this dataset.

| Dataset
Metric | Cora
AUC | Citeseer
AUC | Pubmed
AUC | OGBL-ddi
Hits@20 | OGBL-collab
Hits@50 | OGBL-ppa
Hits@10 | Rank |
|---|---|---|---|---|---|---|---|
| DGI | 90.02 ± 0.80 | 95.53 ± 0.54 | 91.24 ± 0.60 | – | – | – | 11.17 |
| GIC | 93.54 ± 0.60 | 97.04 ± 0.50 | 93.71 ± 0.30 | – | – | – | 9.67 |
| MVGRL | 87.46 ± 0.38 | 88.95 ± 0.66 | 88.36 ± 0.59 | – | – | – | 13.33 |
| BGRL | 87.08 ± 0.24 | 85.82 ± 0.36 | 96.75 ± 0.12 | – | 21.58 ± 1.92 | – | 12.17 |
| GAE | 91.09 ± 0.01 | 90.52 ± 0.04 | 96.40 ± 0.01 | 37.07 ± 5.07 | 44.75 ± 1.07 | 2.52 ± 0.47 | 7.33 |
| GraphSage | 86.33 ± 1.06 | 85.65 ± 2.56 | 89.22 ± 0.87 | 53.90 ± 4.74 | 54.63 ± 1.12 | 1.87 ± 0.67 | 9.00 |
| ARGE | 92.40 ± 0.00 | 91.94 ± 0.00 | 96.81 ± 0.00 | 20.43 ± 4.66 | 28.39 ± 2.51 | 0.41 ± 0.26 | 7.83 |
| GPT-GNN | 92.28 ± 0.31 | 91.36 ± 0.66 | 97.83 ± 0.03 | 37.05 ± 5.96 | 42.41 ± 1.80 | 1.57 ± 0.94 | 6.67 |
| RRL | 88.46 ± 1.85 | 85.47 ± 1.01 | 93.10 ± 0.49 | 16.84 ± 2.23 | 29.88 ± 2.94 | 0.24 ± 0.19 | 10.83 |
| GraphMAE | 89.19 ± 0.00 | 91.20 ± 0.11 | 93.72 ± 0.00 | – | 22.79 ± 1.62 | 0.18 ± 0.28 | 10.92 |
| MaskGAE | **96.66 ± 0.17** | **98.00 ± 0.23** | **98.84 ± 0.04** | 16.25 ± 1.60 | 32.47 ± 0.59 | 0.23 ± 0.04 | 5.00 |
| Bandana | 95.71 ± 0.12 | 96.89 ± 0.21 | 97.26 ± 0.16 | / | 48.67 ± 3.82 | 1.32 ± 1.26 | 4.92 |
| S2GAE-SAGE | 95.05 ± 0.76 | 94.85 ± 0.49 | 97.38 ± 0.17 | 66.00 ± 9.49 | 49.27 ± 0.96 | 1.37 ± 0.38 | 4.67 |
| S2GAE-GCN | 93.52 ± 0.23 | 93.29 ± 0.49 | 98.30 ± 0.12 | 65.91 ± 3.50 | **54.74 ± 1.06** | 3.98 ± 1.33 | 3.83 |
| **Cur-MGAE** | 95.22 ± 0.54 | 95.20 ± 0.31 | 98.43 ± 0.06 | **68.50 ± 5.06** | 52.28 ± 1.35 | **5.96 ± 0.96** | **2.67** |

control the ratio of *easy* edges. For other edges, the probability of the existence of an edge is inversely proportional to the distance of the corresponding nodes' label. In a formulaic representation, the edge formation likelihood between a node 'u' and a node 'v' is dictated by $p_{uc} \propto e^{-|c_u-c_v|}$, where $|c_u - c_v|$ connotes the minimal class interval in a circular arrangement. The *homophily coefficient* is varied in {0.1, 0.5, 0.9} to build three synthetic datasets and test the edge selection strategy on them. Each fabricated graph undergoes a random segmentation into train/validation/test sets with equal number of nodes.

**Learned Edge Selection Curriculum.** Given the aforementioned synthetic datasets, we are able to compare our edge selection curriculum with its intrinsic edge difficulties. We report the proportion of edge selected during the training process in Figure 2, where row indicates different *homophily coefficient*, while columns indicate different *split ratio*, which is a hyperparameter balancing exploration and exploitation in edge selection. From the figures, easy edges are more likely to be selected at the beginning, and difficult edges will be gradually added during the training process. This trend accords with our expectation of the curriculum loss function: before saturation, the number of edges selected in each epoch is positively proportional to the performance of the learned model. The effect of *split ratio* emerges when comparing each column of figures. A smaller *split ratio* will relax the restriction and allow more difficult edges to

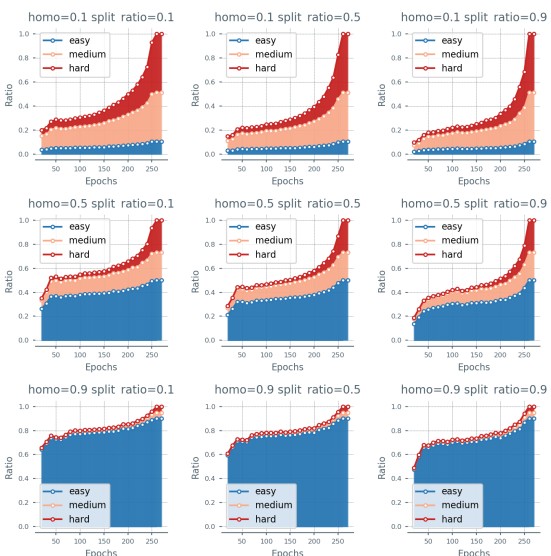

Figure 2: Visualizations of edge selection on different synthetic datasets and edge selection hyperparameters.

be selected at an earlier stage. Since *split ratio* controls the ratio to randomly choose edges, when it is small, more edges are selected randomly, and difficult edges have the chance to be selected. As a result, more edges are selected at the earlier stage of training and the learning processes are sped up, but the tradeoff is the low accuracy. Therefore, we conclude that the tailored structure-aware masking curriculum is effective for learning informative node representations.

### 4.4 ABLATION STUDIES

Furthermore, we conduct ablation studies to discover the most suitable curriculum strategy and verify the effectiveness of the tailored designs. For easier operation, we report results on Cora, Citeseer,

OGBL-ddi and Coauthor-CS in Table 3, while the results on other datasets show similar patterns. The evaluation metric is AUC (%) for link prediction and Accuracy (%) for node classification.

Table 3: Ablation Studies. "Curri." stands for the complexity-guided curriculum masking module, and "CC Dec." means the cross-correlation decoder.

| Datasets | Link Prediction | | | Node Classification | | |
|---|---|---|---|---|---|---|
| | Cora | Citeseer | OGBL-ddi | Cora | Citeseer | Coauthor-CS |
| **Cur-MGAE** | **95.22 ± 0.54** | **95.20 ± 0.31** | **68.50 ± 5.06** | **87.25 ± 0.55** | **74.68 ± 0.37** | **92.69 ± 0.17** |
| w/o Curri. | 92.89 ± 0.40 | 93.66 ± 0.23 | 61.70 ± 9.64 | 86.08 ± 0.15 | 73.92 ± 0.44 | 91.67 ± 0.03 |
| split ratio is 1 | 93.80 ± 1.24 | 92.17 ± 1.13 | 62.90 ± 11.31 | 86.13 ± 0.42 | 74.39 ± 0.17 | 91.58 ± 0.12 |
| split ratio is 0 | 94.12 ± 0.47 | 92.21 ± 0.52 | 62.49 ± 8.97 | 86.93 ± 0.15 | 74.71 ± 0.24 | 91.69 ± 0.02 |
| w/o CC Dec. | 87.43 ± 0.53 | 85.49 ± 0.35 | 22.69 ± 3.65 | 83.05 ± 0.90 | 70.15 ± 0.32 | 90.55 ± 0.24 |
| w/o CC Dec. & Curri. | 87.21 ± 0.69 | 85.18 ± 0.99 | 20.73 ± 1.72 | 82.89 ± 0.11 | 69.09 ± 0.93 | 89.93 ± 0.03 |

**Variant 'w/o Curri.'**. This variant is set to verify the effectiveness of the complexity-guided curriculum masking module (in Section 3.2), which masks the training samples in a easy-to-hard meaningful order. We replaces it with a random mask. The obvious performance drop indicates that it can lead to suboptimal performance by merely treating all data samples for training equally. The test result of this variant demonstrate the effectiveness of the designed structure-aware curriculum.

**Variant 'split ratio is 0'**. For this variant, we set *split ratio* as 0, means that the self-paced mask scheduler (in 3.3) directly chooses a specific number of edges to be masked while does not care about which edges to be masked. This strategy is considered to avoid overfitting, while still leading to model degradation. The performance gap between our model and this variant demonstrates the significance of scheduling edges for training in a meaningful order by the difficulties.

**Variant 'split ratio is 1'**. For this variant, we set *split ratio* as 1, removing the randomness in edge selection of the self-paced mask scheduler (in Section 3.3). Together with the previous variant, these variants are set for verifying the effectiveness of the Self-Paced Mask Scheduler. With the randomness ablated, the edges that have small difficulties in initial training process are always selected by the scheduler, which can potentially lead to overfitting and harm the performances of learned representations. This is shown by the performance drop comparing with our complete model.

**Variant 'w/o CC Dec.'**. This variant is targeting on verifying the effectiveness of the functionality of the specially designed decoder in the structure-aware masked auto-encoder (in Section 3.1). To achieve this, we replaces it with an inner product decoder, while the other modules are kept unchanged. The results of this ablated version drop, indicating that by capturing the common multi-granularity features between connected nodes, the cross-correlation decoder can extract informative representations for reconstruction, which verify the effectiveness of our tailored design.

**Variant 'w/o CC Dec. & Curri.'**. For this variant, we replaces the cross-correlation decoder with inner product decoder and randomly masks the training samples. This ablated version shows the worst performances among all variants, indicating that whichever the decoder is, the proposed structure-aware curriculum strategy plays an important role in learning powerful node representations.

The ablation studies over the key components of our model verify their effectiveness in learning informative node representations. Based on the curriculum we designed, edges with higher difficulty scores can be found. Then stochasticity is utilized to select edges from training sets with the *split ratio* parameter. What's more, the specially designed cross-correlation decoder help with overcoming the flaw contributed by the masking process.

## 5 CONCLUSION

In this paper, we propose a novel curriculum-based graph pretraining strategy utilizing masked graph autoencoder named **Cur-MGAE**. Our proposed method can measure the difficulty of recovering masked edges, which encourages the model to learn from easy parts to difficult parts. The learned graph representations are more informative. Theoretical analysis of the convergence properties is provided in detail. Extensive experimental results on node classification and link prediction tasks demonstrate the effectiveness of the proposed method over state-of-the-art graph SSL baselines.

ETHICS STATEMENT

Our work is built upon publicly available graph datasets that do not contain any sensitive or private information. We have thoroughly reviewed the datasets to ensure that no ethical concerns, such as biased or offensive content, arise from their use. Based on our analysis, we do not anticipate any harmful societal impact or unintended bias resulting from this research. We are committed to ethical standards in research and have ensured that our work aligns with these principles.

REPRODUCIBILITY STATEMENT

To facilitate the reproducibility of our results, we provide detailed information in appendices:

- Appendix C.1: A description of the datasets used and their splits.
- Appendix C.3: Detailed hyperparameter configurations for our proposed method.
- Appendix C.4: Specifications of the hardware and software environments used for experiments.

Additionally, we have listed the detailed pseudocode of training process and will release the complete source code and all necessary scripts to replicate our experiments at the time of publication. This ensures that all aspects of our research can be independently verified.

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

# A   NOTATIONS

Some of the key notations are summarized in Table 4.

Table 4: Notations.

| Notation | Description |
| --- | --- |
| $\mathcal{V}$ | The set of nodes |
| $\mathcal{E}$ | The set of edges |
| $\mathcal{E}_{mask}$ | Masked edges |
| $\mathbf{H}$ | The node representations |
| $\mathbf{A}$ | The adjacency matrix |
| $\mathbf{A}_{mask}$ | The adjacency matrix of the masked edges |
| $\mathbf{A}^{(t)}$ | The adjacency matrix of selected edges at step t |
| $\mathbf{A}_{re}$ | The adjacency matrix of reconstructed edges |
| $\tilde{\mathbf{A}}$ | The predicted adjacency matrix |
| $\mathcal{L}_{all}$ | The overall loss function |
| $\mathcal{L}_{SSL}$ | The loss function for self-supervised learning |
| $\mathcal{L}_{SPCL}$ | The loss function for self-paced curriculum learning |
| $h_v^{(k)}$ | The embedding of node $v$ at the $k$-th layer |
| $N_v$ | The set of direct neighbors of node v |
| COM$(\cdot)$ | The combination function to update the node embedding with the aggregated message |
| AGG$(\cdot)$ | The message aggregation function to aggregate the neighborhood information |
| $N$ | Number of nodes |
| $E$ | Number of edges |
| $d$ | The dimensionality |
| ENC$(\cdot)$ | The encoder |
| DEC$(\cdot)$ | The decoder |
| $h_{e_{v,u}}^{(k)}$ | The final edge representation |
| $K$ | Number of layers |
| $I$ | Neighbors per node for aggregation |
| $\mathcal{K}$ | The number of edges that need to be masked in the training process |
| $g(\cdot)$ | The predicted probability of the presence of an existing edge |
| $S^{(t)}$ | The edge selection matrix at step t |
| $\lambda$ | The regularization coefficient taking control of the number of edges to be selected |
| $\beta$ | The balancing hyper-parameter |
| $\|\cdot\|$ | The $l_2$ norm |
| $w$ | GNN model parameter |
| $\gamma$ | Training transition smoothing regularize coefficient |
| $\omega$ | Mask ratio |

# B   CONVERGENCE GUARANTEES

We have the following convergence guarantees for Algorithm 1:

**Theorem 1 (Avoidance of Saddle Points)** *For a sufficiently large $\gamma$, if the second derivatives of $\mathcal{L}_{SSL}(\boldsymbol{X}, \boldsymbol{A}^{(t-1)}; \boldsymbol{w})$ and $f(\boldsymbol{S}; \lambda)$ are continuous, any bounded sequence $(\boldsymbol{w}^{(t)}, \boldsymbol{S}^{(t)})$ generated by Algorithm 1 with random initialization will not converge to a strict saddle point of F almost surely.*

*Proof. (Avoidance of Saddle Points)* Since the second order derivatives of $\mathcal{L}_{SSL}$ and $f(\boldsymbol{S}; \lambda)$ are continuous. With an assumption of the bounded sequence $(\boldsymbol{w}^{(t)}, \boldsymbol{S}^{(t)})$, the second order derivatives of $\mathcal{L}_{SSL}$ and $f(\boldsymbol{S}; \lambda)$ are bounded. That is to say,

$$max\{||\nabla_{\boldsymbol{w}}^2 \mathcal{L}_{SSL}(\boldsymbol{X}, \boldsymbol{A}^{(t-1)}; \boldsymbol{w})||, ||\nabla_{\boldsymbol{S}}^2 f(\boldsymbol{S}; \lambda)||\} \le p, \tag{6}$$

where $p > 0$ is a constant. Similarly, the second derivative of $\sum_{i,j} S_{ij}||A_{ij} - \tilde{A}_{ij}^{(t)}||$ is bounded, which means $max\{\nabla_{\boldsymbol{w}}^2 \sum_{i,j} S_{ij}||A_{ij} - \tilde{A}_{ij}^{(t)}||, \nabla_{\boldsymbol{S}}^2 \sum_{i,j} S_{ij}||A_{ij} - \tilde{A}_{ij}^{(t)}||\} \le q$, where $q > 0$

is a constant and $\tilde{A}$ is a function of $\boldsymbol{w}$. For this reason, the objective $F$ is bi-smooth, i.e. $max\{||\nabla_{\boldsymbol{w}}^2 F||, ||\nabla_{\boldsymbol{S}}^2 F||\} \leq p + q$, and $F$ satisfies Assumption 4 in (Li et al., 2019b). Thus, from Theorem 10 in (Li et al., 2019b), the second derivative of $F$ is continuous, and for any $\gamma > p + q$, any bounded sequence $(\boldsymbol{w}^{(t)}, \boldsymbol{S}^{(t)})$ generated by Algorithm 1 will not converge to a strict saddle of F almost surely.

**Theorem 2 (Second Order Convergence)** *For a sufficiently large $\gamma$, if the second derivatives of $\mathcal{L}_{SSL}(\boldsymbol{X}, \boldsymbol{A}^{(t-1)}; \boldsymbol{w})$ and $f(\boldsymbol{S}; \lambda)$ are continuous and $\mathcal{L}_{SSL}(\boldsymbol{X}, \boldsymbol{A}^{(t-1)}; \boldsymbol{w})$ and $f(\boldsymbol{S}; \lambda)$ satisfy the Kuradyka-Lojasiewicz (KL) property (Wang et al., 2022b), any bounded sequence $(\boldsymbol{w}^{(t)}, \boldsymbol{S}^{(t)})$ generated by Algorithm 1 with random initialization will almost surely converge to a second-order stationary point of F.*

*Proof. (Second Order Convergence)* In previous proof, $F$ is found to follow Assumption 4 in (Li et al., 2019b). Moreover, since $\mathcal{L}_{SSL}(\boldsymbol{X}, \boldsymbol{A}^{(t-1)}; \boldsymbol{w})$, $f(\boldsymbol{S}; \lambda)$ and $\sum_{i,j} S_{ij}||A_{ij} - \tilde{A}_{ij}^{(t)}||$ satisfy the KL property, F is believed to satisfy the KL property as well. And as proved before, F is continuous. What's more, according to (Wheeden et al., 1977), the continuous differentiability of F implies Lipschitz continuity, which means that the first derivative of F is Lipschitz continuous. As a result, F satisfies Assumption 1 in (Li et al., 2019b). Having F to satisfy both of Assumption 1 and 4, through Corollary 3 in (Li et al., 2019b), for any $\gamma > p + q$, any bounded sequence $(\boldsymbol{w}^{(t)}, \boldsymbol{S}^{(t)})$ generated by Algorithm 1 will almost surely converge to a second-order stationary point of $F$.

## C  ADDITIONAL EXPERIMENTAL DETAILS

### C.1  DATASET STATISTICS

We summarize the statistics of the real-world datasets in the experiments in Table 5. Please note that OGB benchmarks provide train/validation/test split for the standardized experimental protocol. According to the rule explanations on its official website, the validation labels are generally meant for hyper-parameters tuning but are not allowed to be used for model training. The rule also specifically emphasizes that the only exception is the OGBL-collab dataset, where it allows another experimental protocol that validation labels can be used for model training. But in our research, considering that we adopt three OGB link prediction datasets, we adopt the former protocol for OGBL-collab that does not include validation labels in the training process, which is the same as OGBL-ddi and OGBL-ppa.

Table 5: Summary of the dataset statistics.

| Dataset | # Nodes | # Edges | # Features | Train/Val/Test | # Classes |
|---|---|---|---|---|---|
| Cora | 2,708 | 5,429 | 1,433 | 85/5/10 | 7 |
| Citeseer | 3,312 | 4,660 | 3,703 | 85/5/10 | 6 |
| Pubmed | 19,717 | 44,338 | 500 | 85/5/10 | 3 |
| Coauthor-CS | 18,333 | 81,894 | 6,805 | – | 15 |
| Coauthor-Physics | 34,493 | 247,962 | 8,415 | – | 5 |
| ogbn-arxiv | 169,343 | 1,166,243 | 128 | – | 40 |
| OGBL-ddi | 4,267 | 1,334,889 | – | 80/10/10 | – |
| OGBL-collab | 235,868 | 1,285,465 | 128 | 92/4/4 | – |
| OGBL-ppa | 576,289 | 30,326,273 | 58 | 70/20/10 | – |

### C.2  DATASET LICENSE

The datasets included in this work are publicly available as follows:

1. **Plantoid Datasets**: https://github.com/kimiyoung/planetoid/raw/master/data/ with MIT License.

2. **Coauthor Datasets**: https://github.com/shchur/gnn-benchmark/raw/master/data/npz/ with MIT License.

3. **Open Graph Benchmark (OGB)**: https://ogb.stanford.edu.docs/graphprop/ with MIT License.

### C.3 IMPLEMENTATION DETAILS

In our research, we implement our models in PyTorch and use Stochastic Gradient Descent (SGD) for optimization. The number of training epochs is set to 400 for node classification task and 200 for link prediction task. Training patience is 50 steps.

we choose our GNN backbone from GCN (Kipf & Welling, 2017) and GraphSage (Hamilton et al., 2017), while other common message-passing GNNs are also compatible with our model. For large-scale datasets OGBN-arxiv, OGBN-ddi, OGBL-collab, and OGBL-ppa from OGB (Hu et al., 2020a), we set the dimensionality of the representations $d$ as 256 and the number of GNN layers as 3. For the other datasets, we set the dimensionality of the representations $d$ as 128 and the number of GNN layers as 2.

The cross correlation decoder is presented as a two-layer multilayer perceptron (MLP) with ReLU activation function, whose hidden dimension is searched from {128, 256, 512, 1024}. The value for *split ratio* and *mask ratio* are searched in [0, 1] and [0.4, 1] with an interval of 0.1, respectively. And the dropout rate is searched from {0.3, 0.4, 0.5, 0.6}.

For continuously increasing the number of edges that need to be masked in the training process $\mathcal{K}$, the parameter $\lambda$ is increasing during the training process. By our design, $\lambda$ is a function of the training step $t$ and is set as follows:

$$\lambda = \begin{cases} \frac{\lambda_{initial}}{T*\lfloor \frac{2}{3} \rfloor + 1 - t} & if \quad t < T * \lfloor \frac{2}{3} \rfloor, \\ \lambda_{initial} & else. \end{cases} \tag{7}$$

Where $T$ is the total number of epochs and $t$ is the current training step. This function can be replaced depending on the desired curriculum pace.

For the node classification task, we use SVM as the downstream classifier to evaluate the quality of representations output by the baselines and our model. We adopt the 10-fold cross validation accuracy, and report the mean results with standard variation by repeating three times.

For the link prediction task, we randomly sample an equal number of negative samples as the positive samples to compute the AUC score and report the mean results with standard variation after five repeated runs.

The other default hyper-parameters are kept consistent with the work (Tan et al., 2023) for fair comparisons.

### C.4 HARDWARE AND SOFTWARE CONFIGURATION

We conduct the experiments with the following hardware and software configurations:

- Operating System: Ubuntu 20.04.6 LTS
- CPU: Intel(R) Xeon(R) Gold 6348 CPU@2.60GHz
- GPU: NVIDIA GeForce RTX 4090 GPU
- Software: Python 3.8.13; PyTorch 2.0.1; PyTorch Geometric 2.3.1.

## D ADDITIONAL CONTRASTIVE GRAPH SSL METHOD COMPARISON

Since most of the aforementioned baselines are in a generative approach, for a more generalized comparison, node classification results of some contrastive graph self-supervised methods are compared in Table 6.

## E TIME COMPLEXITY ANALYSIS

The time complexity of our model is $O(Ed + Nd^2)$, where $N$, $E$ denotes the total number of nodes and edges in the graphs, $d$ is the dimensionality of the representation. Specifically, our model adopts

Table 6: Node classification accuracy (%) of our proposed method and contrastive baselines. In each column, the boldfaced score denotes the best result of all the methods, and "–" means that the original paper neither considered this dataset nor open-sourced the code publicly available. Rank is the average rank. Our method perform the best in all of the datasets.

| | Cora | Citeseer | Coauthor-Physics | Rank |
|---|---|---|---|---|
| GMI (Peng et al., 2020) | 82.4 ± 0.6 | 71.7 ± 0.2 | – | 12.83 |
| InfoGCL (Xu et al., 2021) | 83.5 ± 0.3 | 73.5 ± 0.4 | – | 9.33 |
| CCA-SSG (Zhang et al., 2021a) | 84.2 ± 0.4 | 73.1 ± 0.3 | 95.38 ± 0.06 | 7.50 |
| GCA-EV (Yu et al., 2023) | – | – | 95.73 ± 0.03 | 10.67 |
| AF-GCL (Wang et al., 2022a) | 83.3 ± 0.1 | 72.1 ± 0.4 | 95.75 ± 0.15 | 7.17 |
| AFGRL (Lee et al., 2022a) | 81.3 ± 0.2 | 68.7 ± 0.3 | 95.69 ± 0.10 | 7.67 |
| SUGRL (Mo et al., 2022) | 83.4 ± 0.5 | 73.0 ± 0.4 | 95.38 ± 0.11 | 6.33 |
| C2F (Zhao et al., 2023) | – | – | 94.09 | 8.33 |
| COSTA-SV (Zhang et al., 2022a) | 84.3 ± 0.3 | 72.8 ± 0.3 | 95.74 ± 0.02 | 5.50 |
| COSTA-MV (Zhang et al., 2022a) | 84.3 ± 0.2 | 72.9 ± 0.3 | 95.60 ± 0.02 | 5.00 |
| IAG (Sun et al., 2023a) | 86.1 | 73.6 | – | 4.00 |
| $S^3$-CL (Ding et al., 2023) | 84.5 ± 0.4 | 74.6 ± 0.4 | – | 3.33 |
| H-GCL (Zhu et al., 2023) | 84.8 ± 0.5 | 74.2 ± 0.3 | – | 2.83 |
| IGCL (Li et al., 2023b) | 79.3 ± 0.1 | 64.2 ± 0.1 | 95.85 ± 0.10 | 2.67 |
| PHASES (Sun et al., 2023b) | – | – | 95.82 ± 0.11 | 2.67 |
| MA-GCL (Gong et al., 2023) | 83.3 ± 0.4 | 73.6 ± 0.1 | – | 2.00 |
| **Cur-MGAE** | **87.3 ± 0.6** | **74.7 ± 0.4** | **95.91 ± 0.05** | **1.00** |

message-passing GNN as the encoder whose time complexity is $O(Ed + Nd^2)$. The time complexity of the decoder and the self-paced mask scheduler is $O(Ed)$, since we calculate the residual for each edge. And the time complexity of the complexity-guided curriculum masking is $O(E)$, because only existing edges are considered to be selected instead of all $N \times N$ potential edges. In comparison, the time complexity of other GNN-based graph representation methods is also $O(Ed + Nd^2)$. Therefore, the time complexity of our proposed model is on par with the existing methods.

In addition, we also conduct the training time comparisons with one competitive baseline S2GAE (Tan et al., 2023) under the same hyper-parameter configurations to verify the efficiency of our model in practice. Specifically, the training epoch is set to 400 and we report the total training time of 3 repeats for node classification task and 5 repeats for link prediction task in the following table. We conduct the experiments with NVIDIA GeForce RTX 4090 GPU and Intel(R) Xeon(R) Gold 6348 CPU @ 2.60GHz. The result is shown in Table 7, we can find that our proposed Cur-MGAE model is also more efficient than S2GAE.

Table 7: Empirical Time Comparisons.

| | Link Pred. | | | Node Class. | | |
|---|---|---|---|---|---|---|
| | Cora | Citeseer | OGBL-ppa | Cora | Citeseer | OGBN-arxiv |
| **Cur-MGAE** | **529.27s** | **336.01s** | **41948.48s** | **157.59s** | **232.50s** | **64502.33s** |
| S2GAE(Tan et al., 2023) | 702.56s | 565.30s | 43161.70s | 304.46s | 534.20s | 65832.35s |

# F  SPACE COMPLEXITY ANALYSIS

As for the space complexity, our model adopts GCN and GraphSAGE as the backbone model, whose space complexity is $O(N \times F + E + \sum_{l=1}^{K} F_{l-1} \times F_l + N \times \sum_{l=1}^{K} F_l)$ and $O(N \times F + E + N \times I^K + \sum_{l=1}^{K} F_{l-1} \times F_l + N \times \sum_{l=1}^{K} F_l)$ respectively, where $N$ is the number of nodes, $F$ is the feature dimension, $E$ is the number of edges, $K$ is the number of layers, $F_{l-1}$ and $F_l$ are the input and output feature dimensions of layer l, and $I$ is neighbors per node for aggregation. In addition to the GNN backbones, our model introduces a complexity-guided curriculum masking and self-paced mask scheduler, whose space complexity are both $O(E)$, which does not induce a higher complexity. Therefore, the space complexity of our proposed model is on par with the existing methods.

In addition, we empirically calculate the number of parameters of the proposed model and baseline models by setting the same embedding dimensionality, which is shown in Table 8. Note that the

parameters are calculated on the Cora dataset, with a hidden dimension of 128. For GraphMAE, the GAT backbone it uses is set to have 4 heads.

| | GraphMAE(Hou et al., 2022) | MaskGAE(Li et al., 2023d) | S2GAE(Tan et al., 2023) | **Cur-MGAE** |
|---|---|---|---|---|
| #Parameters | 419253 | 266370 | 282369 | 291871 |

Table 8: Empirical Space Comparisons.

We can observe that although relatively complex designs are introduced, our method still has a comparable number of parameters with the baseline methods.

## G    SYNTHETIC DATASET VISUALIZATION

A visualization of the synthetically build dataset is given in Figure 3, where points stand for nodes, whose x and y coordinates are derived from intersecting multi-Gaussian distributions. Such nodes are classified into 10 labels based on the node feature, which are represented by the colors. The edge difficulty is correlated with the node labels: edges between nodes with the same label are classified as *easy*, while edges between neighboring nodes and nonadjacent nodes are classified as *medium* and *hard*, respectively.

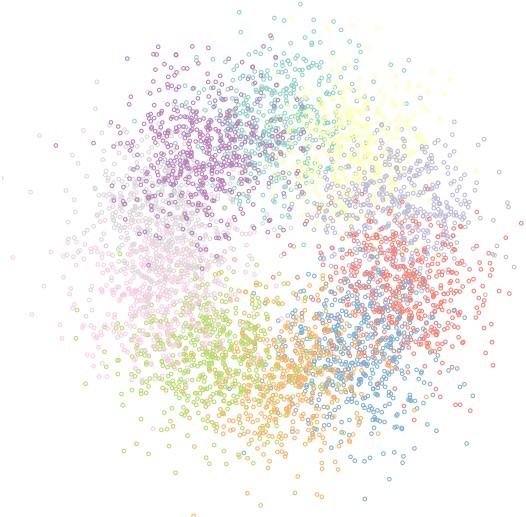

Figure 3: The synthetic dataset.

## H    ADDITIONAL EXPERIMENTS

### H.1    EXPERIMENT RESULTS ON SYNTHETIC DATASETS

Node classification and link prediction experiments are also done on synthetic datasets. Specifically, we consider three of them, whose *homophily coefficient* is 0.1, 0.5, and 0.9, respectively (denoted as Homo = 0.1, 0.5, and 0.9). We adopt two representative baselines in the experiments for comparisons and report the accuracy of link prediction and node classification in the following table.

Table 9: Results on synthetic datasets.

| | Homo=0.1 | | Homo=0.5 | | Homo=0.9 | |
|---|---|---|---|---|---|---|
| | Link Pred. | Node Class. | Link Pred. | Node Class. | Link Pred. | Node Class. |
| **Cur-MGAE** | **52.73 ± 2.56** | **46.79 ± 0.11** | **57.20 ± 0.12** | **82.76 ± 0.67** | **80.97 ± 0.36** | **99.96 ± 0.02** |
| S2GAE(Tan et al., 2023) | 50.04 ± 0.08 | 34.09 ± 0.13 | 51.28 ± 1.27 | 81.90 ± 0.17 | 80.48 ± 0.39 | 98.86 ± 0.00 |
| BGRL(Thakoor et al., 2022) | 51.84 ± 1.88 | 22.93 ± 1.38 | 53.46 ± 0.02 | 40.07 ± 4.40 | 80.68 ± 0.55 | 73.73 ± 0.68 |

From the results above, we can find that our model achieves the best accuracy on all comparisons for both link prediction and node classification, which well verifies the effectiveness of the proposed structure-aware curriculum for learning powerful and informative node representations.

## H.2 HYPERPARAMETER SENSITIVITY

We investigate the sensitivity of some important hyperparameters of our method.

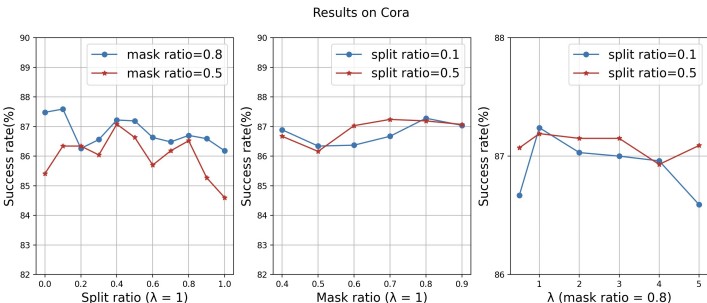

Figure 4: Hyperparameter study on split ratio, mask ratio and initial $\lambda$ by conducting node classification on Cora.

**Effectiveness of split ratio.** *Split ratio* is an important parameter for adding randomness in edged selection for overcoming overfitting. A small *split ratio* represents more edges are selected randomly. Specifically, when *split ratio* is zero, the model selects edges randomly. When *split ratio* is 1.0, the model completely relies on difficulty-based strategy instead. As shown in Figure 4, with a defined mask ratio, instead of 0 or 1, a proper *split ratio* can help with balancing exploitation and exploration, and achieves better result, which is also shown in our ablation studies.

**Effectiveness of mask ratio.** *Mask ratio* defines how many edges can be masked at the maximum. It can be shown in Figure 4 that evidently a smaller *mask ratio* does not help with the increase of model generalizability (e.g. $\omega < 0.7$), while considering other parameters $\omega = 0.8$ is a good value to reach higher performance.

**Effectiveness of initial $\lambda$.** $\lambda$ influences the pace in which our curriculum model increments the number of selected edges. During training, $\lambda$ is increased to force the model mask more and more edges. Thus, the starting point of such parameter will affect the training pace and is of great importance. As shown in Figure 4, for most small datasets like Cora, a relatively small initial $\lambda$ ($\lambda = 1$) is good for digesting the data and shows better performance. However, as this parameter continues decreasing ($\lambda < 1$), the performance will be disturbed because of the lack of data. Due to space limit, the sensitivity experiment on large datasets is not shown here, but in our practice, a larger initial $\lambda$ ($\lambda = 4$) will help the model get more data quickly and perform better.

