# OpenReview forum: "Self-supervised Masked Graph Autoencoder via Structure-aware Curriculum"
_ICLR.cc/2025/Conference — Submitted to ICLR 2025_

### Official Review · Reviewer_i2s5 · 2024-11-02

**Soundness:** 2
**Presentation:** 3
**Contribution:** 2
**Rating:** 3
**Confidence:** 5

**Summary:**

This paper utilizes a masking strategy and an autoencoder structure to model the complexity of edges in order to learn high-quality node representations for downstream tasks.

**Strengths:**

Consider the complexity of edges to optimize graph structures and learn high-quality node representations.

**Weaknesses:**

1. Insufficient motivation. Is edge complexity important in practical problems? What issues exist with the current Mask-GAE structure? Why consider edge complexity in Mask-GAE? What specific problems do you aim to address in Mask-GAE?

2. The scientific definition of edge complexity. Edge complexity is the motivation and starting point of this paper, but the definition of edge complexity in the text is not very scientific. Subtracting the initial structure from the learned structure as a measure of edge complexity—what is the basis for this or its physical significance? Is this definition of complexity generalizable? Could a similar definition be applied to node complexity (initial features minus reconstructed features)? What would be the significance of that?

**Questions:**

Please see Weaknesses.

---

> ### Author Response · Authors · 2024-11-24
> **Response to Reviewer i2s5**
>
> We are grateful for your valuable comments. Please find our responses to each point below.
>
> - **Q1. The motivation on edge complexity.**
>
> R1: Thanks for your comment. We would like to clarify the motivation more clearly as follows:
>
> 1. We focus on a novel and challenging problem, i.e., learning informative representations by designing easy-to-hard mask-reconstruction pretext tasks for masked graph autoencoder. To the best of our knowledge, this problem is not explored by the existing works due to the unique challenges of designing curriculum strategy into graph self-supervised learning.
> 2. Due to the non-Euclidean nature of graph data where nodes are interconnected with each other, most existing curriculum learning works cannot be directly applied to graphs. It also remains a new research direction and is more challenging to design structure-aware curriculum learning strategies.
> 3. We find that for training masked graph autoencoders, the residuals between adjacent values and reconstructed values well quantify the degree that the edges are expected for the GNN encoders at the current stage, where a smaller residual error means the sample should be selected for training at an earlier stage and vice versa. It well aligns with human curriculum learning strategies that can help the model training process [1-6]. Therefore, we propose a tailored difficulty measurer to identify the structural difficulty degree of edges during the masking step. For scheduling edges to be trained, we further propose a novel self-paced scheduler where the intractable discrete optimization of edge selection can be solved by the proposed constrained optimization problem with Lagrange multiplier and also balance exploration and exploitation for training. Such end-to-end model designs are non-trivial.
> 4. We analyze the convergence guarantees of our model. Besides, our model achieves overall performance gains on several synthetic and real-world benchmarks against various baselines. We believe these technical contributions are non-trivial and have potential impacts on the community.
>
> - **Q2. The definition of edge complexity**
>
> R2: Thank you for the question. We have followed your suggestion to add the formal definition to the **edge difficulty** in lines 256-257 in the revised paper as follows: “Edge difficulty is the difficulty score measuring how hard for the current model to predict the edge correctly”.  We follow the idea of curriculum learning, which is also applied in [1-6], to measure the difficulties of each edge and study the edges from easy to hard. To quantify the difficulties of each edge in a given graph, we first use the current model to reconstruct the original graph: $\mathbf{A_{re}} = {\rm DEC}({\rm ENC}(\mathbf{H}, \mathbf{A}))$. The predicted structure $\mathbf{A_{re}}$ reflects the edges that are expected to be inherently drawn from the current model, where the edge probabilities can be viewed as the model’s confidence for the edge occurrence. Intuitively, lower edge confidence indicates that the edge is more complex to be reconstructed by the current model. So we propose to leverage structural complexity to estimate the difficulties of the edge for curriculum learning. Specifically, we calculate the residual errors as: $\mathbf{R} = \mathbf{A} - \mathbf{A_{re}}$. The residual errors measure the distance between the predicted structure and the original structure, which indicates how difficult the edges are for the current model. Consequently, a smaller residual error could be interpreted as an indication of a more pronounced significance of the edge existence. This implies that such edges are relatively easier to learn and comprehend within the context of the model.
>
> We adopted the edge mask-reconstruction as the pretext task so that the definition of node complexity is out of the scope of this work. Nevertheless, we believe a similar idea can be extended on node which can be verified in future work.
>
> References:
>
> [1] Zhao et al., Reinforced curriculum learning on pre-trained neural machine translation models. AAAI, 2020.
>
> [2] Chen et al., Curriculum disentangled recommendation with noisy multifeedback. NeurIPS, 2021.
>
> [3] Li et al., Graph Neural Network with curriculum learning for imbalanced node classification. Neurocomputing, 2024.
>
> [4] Zhang et al., Learning to solve travelling salesman problem with hardness-adaptive curriculum. AAAI, 2022.
>
> [5] Wang et al., CurGraph: Curriculum Learning for Graph Classification. WWW, 2021.
>
> [6] Lan et al., Curriculum multi-negative augmentation for debiased video grounding. AAAI, 2023.
>
> We would greatly appreciate it if you could reconsider your evaluation if some concerns on motivations and the definition are addressed. Thank you!

---

> ### Author Response · Authors · 2024-11-29
> **Follow-Up on Rebuttal Discussion**
>
> Dear Reviewer i2s5,
>
> Thank you for taking the time to review our paper.
>
> As the discussion deadline is approaching, we are eager to hear your feedback on our rebuttal. We hope that our responses have addressed your concerns and would be happy to clarify any further questions.
>
> Looking forward to your valuable input.
>
> Best regards,
> Authors

---

### Official Review · Reviewer_G9un · 2024-11-03

**Soundness:** 2
**Presentation:** 3
**Contribution:** 2
**Rating:** 5
**Confidence:** 4

**Summary:**

To address the issue that existing generative graph self-supervised learning models fails to consider the degrees of difficulties of recovering the masked edges, this paper proposes a curriculum based masked graph autoencoder, Cur-MGAE. Technically, the proposed model equips a difficulty measurer and a self-paced scheduler to modulate the training process of masked graph autoencoders.

**Strengths:**

1. The paper is well-organized and easy to follow.
2. The motivation of considering the difficulties of pretext tasks is reasonable.
3. The paper provides visualization of curriculum learning process.

**Weaknesses:**

1. Technically, although the proposed difficulty measurer appears intuitive, there is a gap in using the residual errors of a graph autoencoder (GAE) to measure the confidence of a masked graph autoencoder (MaskGAE), due to the different self-supervised learning principles of GAE and MaskGAE.
2. The literature review in the paper lacks the latest related works, including but not limited to GraphMAE2 [1], AUG-MAE [2], and Bandana [3]. Notably, AUG-MAE also considers the difficulties of pretext tasks and introduces an easy-to-hard training scheduler. This paper needs to discuss these related works.
3. The resulting edge embeddings in Equation 2 appear to be relatively high-dimensional, so it would be better to provide a comparison of the complexity analysis and model size (number of parameters) between the proposed model and baseline models.

[1] GraphMAE2: A Decoding-Enhanced Masked Self-Supervised Graph Learner, WWW 2023

[2] Rethinking Graph Masked Autoencoders through Alignment and Uniformity, AAAI 2024

[3] Masked Graph Autoencoder with Non-discrete Bandwidths, WWW 2024

**Questions:**

Please refer to the Weaknesses.

---

> ### Author Response · Authors · 2024-11-24
> **Response to Reviewer G9un (Part 1/2)**
>
> We thank the reviewer for the insightful comments. Please find our responses to each point below.
>
> - **Q1. There is a gap in using the residual errors of GAE to measure the confidence of a Masked graph autoencoder.**
>
> R1: Thank you for this comment. We would like to clarify the motivation using the residual errors of GAE as the difficulty measurer is that at the beginning phase, we could not determine which edges should be masked, and the function of the difficulty measurer is to find such edges. For this reason, Masked graph autoencoder can't be applied at this stage. Specifically, to quantify the difficulties of each edge in a given graph, we first use the current model to reconstruct the original graph: $\mathbf{A_{re}} = {\rm DEC}({\rm ENC}(\mathbf{H}, \mathbf{A}))$. The predicted structure $\mathbf{A_{re}}$ reflects the edges that are expected to be inherently drawn from the current model, where the edge probabilities can be viewed as the model’s confidence for the edge occurrence. Intuitively, lower edge confidence indicates that the edge is more complex to be reconstructed by the current model. So we propose to leverage structural complexity to estimate the difficulties of the edge for curriculum learning. Specifically, we calculate the residual errors as: $\mathbf{R} = \mathbf{A} - \mathbf{A_{re}}$. The residual errors measure the differences between the predicted structure and the original structure, which indicate how difficult the edges are for the current model. Consequently, a smaller residual error could be interpreted as an indication of a more pronounced significance of the edge existence. This implies that such edges are relatively easier to learn and comprehend within the context of the model. We have revised the expression in lines 256-268 for better clarification in the revised paper.
>
> - **Q2. More related works need to be discussed, e.g., GraphMAE2, AUG-MAE, and Bandana**
>
> R2: Thank you for this suggestion. In our experiments, we compared with some representative baselines in the field. To further compare our method with more updated methods as you suggested, we have adopted 3 more recent methods, i.e., Bandana [1], GraphMAE2 [2], and AUG-MAE [3]. Note that  Bandana [1] proposes a novel framework for masked graph autoencoder that utilizes non-discrete edge masks, referred to as "bandwidths". Unlike discrete masking, these bandwidths are sampled from a continuous and dispersive probability distribution, allowing for a more nuanced restriction of message propagation through each edge, which maintains the integrity of the graph's topology during training, ensuring that long-range information can flow unimpeded and enhancing the model's ability to learn fine-grained neighborhood structures. GraphMAE2 [2] is a self-supervised learning framework designed to enhance graph representation learning by increasing the model robustness. Compared with its predecessor GraphMAE [4], the multi-view random re-mask decoding and latent representation prediction modules are designed to help with noisy and indistinguishable input features. AUG-MAE [3] targets on node and graph classification tasks, introducing an easy-to-hard adversarial masking approach that generates challenging samples for the model to reconstruct. This strategy enhances the model's ability to align representations of similar nodes, leading to more accurate and robust embeddings. The experimental results are shown in the below table. We can observe that our Cur-MGAE consistently achieves the best performance compared with these additional baselines. Noted that “/” indicates that the current method can’t be directly used on the datasets that do not have node features, and GraphMAE2 and AUG-MAE are omitted in the link prediction task since they are node or graph classification methods.
>
> | Node Classification | Cora | Citeseer | PubMed | Coauthor-CS | Coauthor-Physics | OGBN-arxiv |
> | --- | --- | --- | --- | --- | --- | --- |
> | GraphMAE2 [2] | 84.50±0.60 | 73.40±0.30 | 81.40±0.50 | 92.13±0.12 | 95.44±0.08 | 71.89±0.03 |
> | AUG-MAE [3] | 84.30±0.40 | 73.20±0.40  | 81.40±0.40 | 92.15±0.22 | 95.34±0.60 | 71.90±0.20 |
> | Bandana [1] | 84.62±0.37 | 73.60±0.16 | 83.53±0.51 | **93.10±0.05** | 95.57±0.04 | 71.09±0.24 |
> | Cur-MGAE | **87.25±0.55**  | **74.68±0.37**  | **85.86±0.14**  | 92.69±0.17  | **95.91±0.05**  | **73.00±0.06** |
>
> | Link Prediction | Cora(AUC) | Citeseer(AUC) | PubMed(AUC) | OGBL-ddi(Hits@20) | OGBL-collab(Hits@50) | OGBL-ppa(Hits@10) |
> | --- | --- | --- | --- | --- | --- | --- |
> | Bandana [1] | **95.71±0.12** | **96.89±0.21** | 97.26±0.16 | / | 48.67±3.82 | 1.32±1.26 |
> | Cur-MGAE | 95.22±0.54  | 95.20±0.31  | **98.43±0.06**  | **68.50±5.06**  | **52.28±1.35**  | **5.96±0.96** |
>
> We have added the above results in Table 1 and Table 2 in the revised paper.

---

> ### Author Response · Authors · 2024-11-24
> **Response to Reviewer G9un (Part 2/2)**
>
> - **Q3. The number of parameters of the proposed model and the baselines.**
>
> R3: Thank you for this comment. Following your suggestion, we have first analyzed the space complexity of our method and baselines as follows. “Ours model adopts GCN and GraphSAGE as the backbone model, whose space complexity is $O(N \times F + E + \sum^{K}\_{l=1}F\_{l-1} \times F\_{l} + N \times \sum^K\_{l=1}F\_l)$ and $O(N \times F + E + N \times I^K +  \sum^{K}\_{l=1}F\_{l-1} \times F\_{l} + N \times \sum^K\_{l=1}F\_l)$ respectively, where $N$ is the number of nodes, $F$ is the feature dimension, $E$ is the number of edges, $K$ is the number of layers, $F\_{l-1}$ and $F\_l$ are the input and output feature dimensions of layer $l$, and $I$ is neighbors per node for aggregation. In addition to the GNN backbones, our model introduces a complexity-guided curriculum masking and self-paced mask scheduler, whose space complexity are both $O(E)$, which does not induce a higher complexity. Therefore, the space complexity of our proposed model is on par with the existing methods”
>
> In addition, we empirically calculate the number of parameters of the proposed model and baseline models by setting the same embedding dimensionality, which is shown in the following table. Note that the parameters are calculated on the Cora dataset, with a hidden dimension of 128. For GraphMAE, the GAT it uses is set to have 4 heads.
>
> |  | GraphMAE | MaskGAE | S2GAE | Cur-MGAE |
> | --- | --- | --- | --- | --- |
> | #Parameters | 419253 | 266370 | 282369 | 291871 |
>
> We can observe that although relatively complex designs are introduced, our method still has a comparable number of parameters with the baseline methods. We have added the detailed results and discussion in lines 1068-1089 in the revised paper.
>
> References:
>
> [1] Zhao et al., Masked Graph Autoencoder with Non-discrete Bandwidths. WWW, 2024.
>
> [2] Hou et al., GraphMAE2: A Decoding-Enhanced Masked Self-Supervised Graph Learner. WWW, 2023.
>
> [3] Wang et al., Rethinking Graph Masked Autoencoders through Alignment and Uniformity. AAAI, 2024.
>
> [4] Hou et al., GraphMAE: Self-Supervised Masked Graph Autoencoders. KDD, 2022.
>
> We hope our response can effectively address your initial concerns regarding method design, related works, and complexity. We sincerely appreciate your reconsideration of our work in light of our explanations.

---

> ### Author Response · Authors · 2024-11-29
> **Follow-Up on Rebuttal Discussion**
>
> Dear Reviewer G9un,
>
> Thank you for taking the time to review our paper.
>
> As the discussion deadline is approaching, we are eager to hear your feedback on our rebuttal. We hope that our responses have addressed your concerns and would be happy to clarify any further questions.
>
> Looking forward to your valuable input.
>
> Best regards,
>
> Authors

---

> > ### Comment · Reviewer_G9un · 2024-12-03
> >
> > Thank you for your detailed response. Most of my concerns have been addressed. However, the motivation for using the residual errors of the graph autoencoder (GAE) as the difficulty measurer for the masked graph autoencoder (MaskGAE) is still not entirely convincing. Therefore, I have decided to maintain my score.

---

### Official Review · Reviewer_bEWC · 2024-11-03

**Soundness:** 2
**Presentation:** 3
**Contribution:** 2
**Rating:** 5
**Confidence:** 4

**Summary:**

This paper presents a novel approach to self-supervised learning (SSL) for graph-structured data, specifically through an enhanced masked graph autoencoder. While existing methods have achieved success in recovering masked parts of graph data, they overlook the varying difficulties associated with masking edges, which can lead to suboptimal representations of nodes.
To address this issue, the authors introduce a curriculum-based self-supervised masked graph autoencoder that effectively captures the inherent difficulty levels of edge recovery. The approach involves the following parts:
1. Difficulty Measurer: A mechanism is proposed to assess the structural difficulty of edges during the masking process.

2. Self-Paced Scheduler: This component determines the sequence in which edges are masked, allowing the graph encoder to gradually progress from easier to more difficult tasks.

3. Adaptive Reconstruction Tasks: The masked edges are incrementally included in the reconstruction tasks, which fosters the learning of more informative node representations.

Extensive experiments on various real-world datasets for node classification and link prediction demonstrate that the proposed method outperforms existing state-of-the-art baselines in graph self-supervised learning. This work is notable as it is the first to apply a curriculum learning strategy specifically to masked graph autoencoders.

**Strengths:**

1. The introduction of a curriculum-based strategy for masked graph autoencoders is a potential solution to the field of self-supervised learning on graph data. This approach is novel and interesting. This idea is innovative and may provide insights to other researchers.

2. By taking into account the different levels of difficulty in masking edges, the proposed method seeks to enhance how node representations are learned. This adds an important improvement to traditional approaches by recognizing that some masked parts are harder to reconstruct than others.

**Weaknesses:**

1. Scalability Concerns: Depending on how the difficulty measurer and self-paced scheduler are implemented, there may be questions regarding scalability to very large graphs, where edge masking and reconstruction could become computationally intensive.

2. Insufficient baselines: Although the paper experiments with multiple datasets, the chosen GNN baselines are relatively outdated and perform exceptionally well. For instance, the OGBN-Arxiv leaderboard features numerous models, such as GLEM, GIANT, and SimTeg, that have surpassed the baseline. Furthermore, well-established frameworks like GAT are not included in the comparison, which raises questions about the effectiveness of the proposed model and its competitive standing in the current landscape of graph neural networks.

3. Theoretical Foundation: Although the paper introduces a difficulty level for edge masking, there is a lack of robust theoretical justification for how these difficulty levels are defined. This ambiguity raises concerns about the validity of the proposed curriculum approach, as it is not clear why these defined levels are beneficial compared to traditional methods. A stronger theoretical foundation would enhance the contribution of the work and clarify the rationale behind the chosen difficulty metrics.

4. Complexity of Implementation: The introduction of a curriculum-based approach may complicate the model's architecture and the overall training process, which could pose challenges in terms of reproducibility and practical deployment.

**Questions:**

1. Why did you choose the specific GNN baselines included in your experiments? Were there considerations for including more recent or competitive models, such as GAT or models from the OGBN-Arxiv leaderboard?

2. What theoretical principles support the effectiveness of the curriculum-based approach for masked graph autoencoders?

3. How did you determine the criteria for evaluating the difficulty levels of masking edges? Can you provide more details on the methodology behind this assessment?

---

> ### Author Response · Authors · 2024-11-24
> **Response to Reviewer bEWC (Part 1/3)**
>
> Thank you for your efforts and insightful comments. We have provided detailed responses to address each of your concerns below.
>
> - **Q1. The scalability concerns on the difficulty measurer and self-paced scheduler.**
>
> R1: Thank you for this question. We analyzed the time complexity of our method in section E in the Appendix. Following your suggestion, we would like to discuss in more detail on the scalability as follows. The time complexity of our model is on par with the baselines, i.e., $O(Ed+Nd^2)$, where $N$, $E$ denotes the total number of nodes and edges in the graphs, $d$ is the dimensionality of the representation. Specifically, the time complexity of the self-paced mask scheduler is $O(Ed)$, since we calculate the residual for each edge. The time complexity of the complexity-guided curriculum masking is $O(E)$ because only existing edges are considered to be selected instead of all $N\times N$ potential edges. In addition, we adopt message-passing GNN as the encoder whose time complexity is $O(Ed+Nd^2)$. Overall, the computational efficiency of our method is promising.
>
> Besides the results of the empirical time comparisons in Table 7 in the Appendix, I also conducted experiments on larger graphs OGBN-arxiv and OGBL-ppa. The results are shown in the following table. Noted that we report the total training time of 3 repeats for the node classification task and 5 repeats for the link prediction task, which accord with the experimental setting in Table 7.
>
> |  | OGBN-arxiv (s) | OGBL-ppa (s) |
> | --- | --- | --- |
> | S2GAE | 65832.35 | 43161.70 |
> | Cur-MGAE | 64502.33 | 41948.48 |
>
> We can also observe the good efficiency of our method, although it consists of the designs of the edge masking and reconstruction. We have added the detailed discussions into section E in the Appendix. Please refer to the revised paper for more details.
>
> - **Q2. The baselines are insufficient.**
>
> R2. Thanks for the comment.  In our experiments, we compared with some representative baselines in the field. We followed the well-established evaluation setting in the literature [1] which is using SVM as the downstream classifier to evaluate the quality of representations output by the baselines and our model. So the results of our method are different from the GLEM, GIANT, SimTeg, and the methods on the leaderboard because the evaluation setting is different. We also noticed that in the literature, these methods are also not considered as the baselines in the related works [1-6].
>
> Nevertheless, we have followed your suggestions to add 3 more recent methods, i.e., Bandana [3], GraphMAE2 [4], and AUG-MAE [5]. Note that  Bandana [1] proposes a novel framework for masked graph autoencoder that utilizes non-discrete edge masks, referred to as "bandwidths". Unlike discrete masking, these bandwidths are sampled from a continuous and dispersive probability distribution, allowing for a more nuanced restriction of message propagation through each edge, which maintains the integrity of the graph's topology during training, ensuring that long-range information can flow unimpeded and enhancing the model's ability to learn fine-grained neighborhood structures. GraphMAE2 [4] is a self-supervised learning framework designed to enhance graph representation learning by increasing the model robustness. Compared with its predecessor GraphMAE [6], the multi-view random re-mask decoding and latent representation prediction modules are designed to help with noisy and indistinguishable input features. AUG-MAE [5] targets on node and graph classification tasks, introducing an easy-to-hard adversarial masking approach that generates challenging samples for the model to reconstruct. This strategy enhances the model's ability to align representations of similar nodes, leading to more accurate and robust embeddings. The experimental results are shown in the below table. We can observe that our Cur-MGAE consistently achieves the best performance compared with these additional baselines. Noted that “/” indicates that the current method can’t be directly used on the datasets that do not have node features, and GraphMAE2 and AUG-MAE are omitted in the link prediction task since they are node or graph classification methods.

---

> ### Author Response · Authors · 2024-11-24
> **Response to Reviewer bEWC (Part 2/3)**
>
> | Node Classification | Cora | Citeseer | PubMed | Coauthor-CS | Coauthor-Physics | OGBN-arxiv |
> | --- | --- | --- | --- | --- | --- | --- |
> | GraphMAE2 [4] | 84.50±0.60 | 73.40±0.30 | 81.40±0.50 | 92.13±0.12 | 95.44±0.08 | 71.89±0.03 |
> | AUG-MAE [5] | 84.30±0.40 | 73.20±0.40  | 81.40±0.40 | 92.15±0.22 | 95.34±0.60 | 71.90±0.20 |
> | Bandana [3] | 84.62±0.37 | 73.60±0.16 | 83.53±0.51 | **93.10±0.05** | 95.57±0.04 | 71.09±0.24 |
> | Cur-MGAE | **87.25±0.55**  | **74.68±0.37**  | **85.86±0.14**  | 92.69±0.17  | **95.91±0.05**  | **73.00±0.06** |
>
> | Link Prediction | Cora(AUC) | Citeseer(AUC) | PubMed(AUC) | OGBL-ddi(Hits@20) | OGBL-collab(Hits@50) | OGBL-ppa(Hits@10) |
> | --- | --- | --- | --- | --- | --- | --- |
> | Bandana [3] | **95.71±0.12** | **96.89±0.21** | 97.26±0.16 | / | 48.67±3.82 | 1.32±1.26 |
> | Cur-MGAE | 95.22±0.54  | 95.20±0.31  | **98.43±0.06**  | **68.50±5.06**  | **52.28±1.35**  | **5.96±0.96** |
>
> We have added the above results in Table 1 and Table 2 in the revised paper.
>
> - **Q3. The theoretical principles supporting why these defined levels in curriculum-based approach are beneficial compared to traditional methods?**
>
> R3: Thank you for this comment.  We provided theoretical analyses for the convergence properties of our method, which is the main aspect to show the effectiveness of the curriculum-based approach following the literature [7-12] and supports our model design. Besides, we empirically verify that adopting curriculum learning strategies can achieve better experimental results.
>
> - **Q4. Complexity of Implementation**
>
> R4: Thank you for this comment. We would like to clarify that the curriculum learning strategies in self-supervised masked graph autoencoder are one of the main contributions of our method, which is able to capture and leverage the underlying degree of difficulties of data dependencies hidden in edges and design better mask-reconstruction pretext tasks for learning informative representations. Its effectiveness is verified in the node classification tasks (Table 1) and link prediction (Table 2) tasks. In addition, we analyzed the time and space complexity of our method is comparable with the baseline methods (in sections E and F in Appendix), which does not introduce a high complexity of implementation.
>
> - **Q5. More recent or competitive baselines.**
>
> R5: Thank you for the question. We followed the related works [1-3] to choose the GCN and GraphSage as our backbones for a fair comparison. Following your suggestion, 3 more recent baselines where GraphMAE2 and AUG-MAE use GAT as the default backbone in their original paper are compared. The results in Q2 show that our method still outperforms the baselines that use GAT as the backbone.

---

> ### Author Response · Authors · 2024-11-24
> **Response to Reviewer bEWC (Part 3/3)**
>
> - **Q6.  Theoretical principles support the effectiveness of the curriculum-based approach**
>
> R6: Thank you for this question. We hope the response to Q3 might address your concern about the theoretical guarantees.
>
> - **Q7.  How to determine the criteria for evaluating the difficulty levels of masking edges?**
>
> R7: We have revised the expressions for better clarification on how to determine the criteria for evaluating the difficulty levels of masking edges as follows “To quantify the difficulties of each edge in a given graph, we first use the current model to reconstruct the original graph: $\mathbf{A_{re}} = {\rm DEC}({\rm ENC}(\mathbf{H}, \mathbf{A}))$. The predicted structure $\mathbf{A_{re}}$ reflects the edges that are expected to be inherently drawn from the current model, where the edge probabilities can be viewed as the model’s confidence for the edge occurrence. Intuitively, lower edge confidence indicates that the edge is more complex to be reconstructed by the current model. So we propose to leverage structural complexity to estimate the hardness of the edge for curriculum learning”. Please refer to lines 256-268 in the revised paper for details.
>
> Reference:
>
> [1] Tan et al., S2GAE: Self-Supervised Graph Autoencoders Are Generalizable Learners with Graph Masking. WSDM, 2023.
>
> [2] Li et al., What’s Behind the Mask: Understanding Masked Graph Modeling for Graph Autoencoders. KDD, 2023.
>
> [3] Zhao et al., Masked Graph Autoencoder with Non-discrete Bandwidths. WWW, 2024.
>
> [4] Hou et al., GraphMAE2: A Decoding-Enhanced Masked Self-Supervised Graph Learner. WWW, 2023.
>
> [5] Wang et al., Rethinking Graph Masked Autoencoders through Alignment and Uniformity. AAAI, 2024.
>
> [6] Hou et al., GraphMAE: Self-Supervised Masked Graph Autoencoders. KDD, 2022.
>
> [7] Zhao et al., Reinforced curriculum learning on pre-trained neural machine translation models. AAAI, 2020.
>
> [8] Chen et al., Curriculum disentangled recommendation with noisy multifeedback. NeurIPS, 2021.
>
> [9] Li et al., Graph Neural Network with curriculum learning for imbalanced node classification. Neurocomputing, 2024.
>
> [10] Zhang et al., Learning to solve travelling salesman problem with hardness-adaptive curriculum. AAAI, 2022.
>
> [11] Wang et al., CurGraph: Curriculum Learning for Graph Classification. WWW, 2021.
>
> [12] Lan et al., Curriculum multi-negative augmentation for debiased video grounding. AAAI, 2023.
>
> We hope our response can effectively address your initial concerns on scalability, baselines, theories, and complexity. We sincerely appreciate your reconsideration of our work in light of our explanations. We would be glad to engage further if there are any additional points we may have missed. Thank you!

---

> > ### Comment · Reviewer_bEWC · 2024-11-27
> >
> > Thank you for your efforts in providing additional experiments and explanations. They have addressed most of my concerns. However, I still feel that the theoretical support is not strong enough. The method relies heavily on the assumption that lower edge confidence directly correlates with structural complexity. However, this may not always hold true. There could be edges that are inherently complex but still have high confidence. While empirically useful,  it lacks a solid theoretical foundation. There are many existing masked-based methods, and the proposed method may not be well generalized across different types of graphs or applications. Thus, I will keep my score as "marginally below the acceptance threshold".

---

### Official Review · Reviewer_LL8M · 2024-11-04

**Soundness:** 3
**Presentation:** 3
**Contribution:** 2
**Rating:** 6
**Confidence:** 5

**Summary:**

This paper presents Cur-MGAE, a new self-supervised approach for graphs that brings curriculum learning into masked graph autoencoders. The method uses a structure-aware design to recover masked edges in an easy-to-hard order, guided by reconstruction errors and a self-paced scheduler. Theoretical analysis shows the optimization converges, and experiments on node classification and link prediction tasks demonstrate Cur-MGAE’s strong performance, showing how well this approach captures meaningful node representations.

**Strengths:**

1. The paper introduces complexity-guided curriculum masking and self-paced mask scheduling as thoughtful strategies to integrate curriculum learning into graph masked autoencoder. These methods aim to structure the learning process in a way that aligns task difficulty with model capacity, potentially improving performance by enabling the model to handle simpler patterns before moving on to more complex structures in the graph. This structured approach provides a clear framework for applying curriculum learning principles effectively within graph masked autoencoder.
2. The paper does a great job of providing a detailed theoretical analysis of the convergence properties for the alternating optimization algorithm used in Cur-MGAE.

**Weaknesses:**

1. There are some typos in this paper. In line 359 and 363, the reference for PubMed is missing. Also, I believe the dataset name is misspelling, the “citepseer” should be “CiteSeer”?
2. The selected baselines are bit too old. I suggest the authors select more updated methods as your baselines [1].
3. It would be helpful if the paper discussed Cur-MGAE's computational efficiency a bit more. With the self-paced mask scheduler and iterative edge selection, there might be extra computational overhead. When dealing large graphs, will these factors affect the efficiency of Cur-MGAE?

[1] Hou, Zhenyu, et al. "Graphmae2: A decoding-enhanced masked self-supervised graph learner." Proceedings of the ACM Web Conference 2023. 2023.

**Questions:**

See weakness.

---

> ### Author Response · Authors · 2024-11-24
> **Response to Reviewer LL8M (Part 1/2)**
>
> We sincerely appreciate the reviewer for the valuable positive feedback. We addressed all the comments. Please kindly find the detailed responses below.
>
> - **Q1. Some typos in the paper.**
>
> R1: Thank you for the comment. We have followed your suggestion to add the reference for PubMed in lines 363 and 367, and revise the “citepseer” into “CiteSeer”.
>
> - **Q2. More recent baselines should be compared.**
>
> R2: Thank you for this suggestion. In our experiments, we compared with some representative baselines in the field. To further compare our method with more updated methods as you suggested, we have adopted 3 more recent methods, i.e., Bandana [1], GraphMAE2 [2], and AUG-MAE [3]. Note that  Bandana [1] proposes a novel framework for masked graph autoencoder that utilizes non-discrete edge masks, referred to as "bandwidths". Unlike discrete masking, these bandwidths are sampled from a continuous and dispersive probability distribution, allowing for a more nuanced restriction of message propagation through each edge, which maintains the integrity of the graph's topology during training, ensuring that long-range information can flow unimpeded and enhancing the model's ability to learn fine-grained neighborhood structures. GraphMAE2 [2] is a self-supervised learning framework designed to enhance graph representation learning by increasing the model robustness. Compared with its predecessor GraphMAE [4], the multi-view random re-mask decoding and latent representation prediction modules are designed to help with noisy and indistinguishable input features. AUG-MAE [3] targets on node and graph classification tasks, introducing an easy-to-hard adversarial masking approach that generates challenging samples for the model to reconstruct. This strategy enhances the model's ability to align representations of similar nodes, leading to more accurate and robust embeddings. The new experimental results are shown in the table below. We can observe that our Cur-MGAE consistently achieves the best performance compared with these additional baselines. Noted that “/” indicates that the current method can’t be directly used on the datasets that do not have node features, and GraphMAE2 and AUG-MAE are omitted in the link prediction task since they are node or graph classification methods.
> | Node Classification | Cora | Citeseer | PubMed | Coauthor-CS | Coauthor-Physics | OGBN-arxiv |
> | --- | --- | --- | --- | --- | --- | --- |
> | GraphMAE2 [2] | 84.50±0.60 | 73.40±0.30 | 81.40±0.50 | 92.13±0.12 | 95.44±0.08 | 71.89±0.03 |
> | AUG-MAE [3] | 84.30±0.40 | 73.20±0.40  | 81.40±0.40 | 92.15±0.22 | 95.34±0.60 | 71.90±0.20 |
> | Bandana [1] | 84.62±0.37 | 73.60±0.16 | 83.53±0.51 | **93.10±0.05** | 95.57±0.04 | 71.09±0.24 |
> | Cur-MGAE | **87.25±0.55**  | **74.68±0.37**  | **85.86±0.14**  | 92.69±0.17  | **95.91±0.05**  | **73.00±0.06** |
>
> | Link Prediction | Cora(AUC) | Citeseer(AUC) | PubMed(AUC) | OGBL-ddi(Hits@20) | OGBL-collab(Hits@50) | OGBL-ppa(Hits@10) |
> | --- | --- | --- | --- | --- | --- | --- |
> | Bandana [1] | **95.71±0.12** | **96.89±0.21** | 97.26±0.16 | / | 48.67±3.82 | 1.32±1.26 |
> | Cur-MGAE | 95.22±0.54  | 95.20±0.31  | **98.43±0.06**  | **68.50±5.06**  | **52.28±1.35**  | **5.96±0.96** |
>
> We have added the above results in Table 1 and Table 2 in the revised paper.

---

> ### Author Response · Authors · 2024-11-24
> **Response to Reviewer LL8M (Part 2/2)**
>
> - **Q3. More discussions on computational efficiency. Will the self-paced mask scheduler and iterative edge selection induce a higher computational cost?**
>
> R3: Thank you for this question. We would like to clarify that the time complexity of our model is on par with the baselines, i.e., $O(Ed+Nd^2)$, where $N$, $E$ denotes the total number of nodes and edges in the graphs, $d$ is the dimensionality of the representation. Specifically, the time complexity of the self-paced mask scheduler is $O(Ed)$, since we calculate the residual for each edge. The time complexity of the complexity-guided curriculum masking is $O(E)$ because only existing edges are considered to be selected instead of all $N\times N$ potential edges. In addition, we adopt message-passing GNN as the encoder whose time complexity is $O(Ed+Nd^2)$. Overall, the computational efficiency of our method is promising.
>
> Besides the results of the empirical time comparisons in Table 7 in the Appendix, we have also conducted experiments on larger graphs OGBN-arxiv and OGBL-ppa. The results are shown in the following table. Noted that we report the total training time of 3 repeats for the node classification task and 5 repeats for the link prediction task, which accord with the experimental setting in Table 7.
>
> |  | OGBN-arxiv (s) | OGBL-ppa (s) |
> | --- | --- | --- |
> | S2GAE | 65832.35 | 43161.70 |
> | Cur-MGAE | 64502.33 | 41948.48 |
>
> We can also observe the good efficiency of our method. Please refer to the line 1023-1065 in the revised paper for more details.
>
> Reference:
>
> [1] Zhao et al., Masked Graph Autoencoder with Non-discrete Bandwidths. WWW, 2024.
>
> [2] Hou et al., GraphMAE2: A Decoding-Enhanced Masked Self-Supervised Graph Learner. WWW, 2023.
>
> [3] Wang et al., Rethinking Graph Masked Autoencoders through Alignment and Uniformity. AAAI, 2024.
>
> [4] Hou et al., GraphMAE: Self-Supervised Masked Graph Autoencoders. KDD, 2022.
>
>
> We sincerely appreciate your positive evaluation of our work and have conducted additional experiments based on your valuable suggestions. We would very much appreciate it if you could reconsider your evaluation if some concerns are addressed. Thank you very much!

---

> ### Author Response · Authors · 2024-11-29
> **Follow-Up on Rebuttal Discussion**
>
> Dear Reviewer LL8M,
>
> Thank you for taking the time to review our paper.
>
> As the discussion deadline is approaching, we are eager to hear your feedback on our rebuttal. We hope that our responses have addressed your concerns and would be happy to clarify any further questions.
>
> Looking forward to your valuable input.
>
> Best regards,
>
> Authors

---

### Author Response · Authors · 2024-11-24
**General Response**

We sincerely appreciate the reviewers' efforts and valuable feedback on our paper.

|     | Reviewer LL8M | Reviewer bEWC | Reviewer G9un | Reviewer i2s5 | Action |
| --- | --- | --- | --- | --- | --- |
| Motivation | “provides a clear framework for applying curriculum learning principles effectively within graph masked autoencoder.” | “is a potential solution to the field of self-supervised learning on graph data.” | “The motivation of considering the difficulties of pretext tasks is reasonable” | “Is edge complexity important in practical problems?” | Re Reviewer i2s5: `We further revised some expressions to clarify the motivations more clearly` |
| Novelty | “presents Cur-MGAE, a new self-supervised approach for graphs that brings curriculum learning into masked graph autoencoders.” | “This approach is novel and interesting. This idea is innovative and may provide insights to other researchers.” | N/A | N/A |   `N/A` |
| Presentation | “some typos” | N/A | “The paper is well-organized and easy to follow”, “provides visualization of curriculum learning process” | N/A | Re Reviewer LL8M: `We fixed the typos` |
| Theoretical Analysis | “does a great job of providing a detailed theoretical analysis of the convergence properties for the alternating optimization algorithm” | N/A | N/A | N/A | `N/A` |
| Efficiency | “It would be helpful if the paper discussed Cur-MGAE's computational efficiency a bit more” | “there may be questions regarding scalability to very large graphs” | “it would be better to provide a comparison of the complexity analysis and model size” | N/A | `We theoretically and empirically analyzed the time and space complexity of our method to show the good efficiency.` |
| Extended Evaluation | “suggest the authors select more updated methods as your baselines” | “Were there considerations for including more recent or competitive models” | “The literature review in the paper lacks the latest related works” | N/A | `We added more recent baselines to compare with our method and updated the result table.` |
| Effectiveness | “potentially improving performance by enabling the model to handle simpler patterns before moving on to more complex structures in the graph” | “This adds an important improvement to traditional approaches by recognizing that some masked parts are harder to reconstruct than others” | N/A | “Consider the complexity of edges to optimize graph structures and learn high-quality node representations” | `N/A` |

## **Summary**

We have addressed all the comments point by point, primarily on updating the experiments by adding comparisons with recent methods and revising some expressions for better clarification.

We sincerely appreciate the reviewers' insightful suggestions, which have significantly enhanced the quality of our work. We hope our responses have clarified any uncertainties and effectively addressed the concerns raised.

Please let us know if our revisions have met your expectations or if there are any additional questions or issues we can address!

---

### Meta-Review · Area_Chair_ATDg · 2024-12-20

**Metareview:**

The paper presents Cur-MGAE, a new self-supervised approach for graphs that brings curriculum learning into masked graph autoencoders. The proposed method is interesting and reasonable. Cur-MGAE outperforms the baseline methods. The paper presentation is good. Some key issues raised by reviewers need to be addressed, including insufficient baseline methods, a lack of important related works and robust theoretical justification, and unconvincing motivation of edge complexity. Reviewers reach a consensus and are generally negative about this work.

**Additional Comments On Reviewer Discussion:**

Reviewers reach a consensus and are generally negative about this work. The reviewer who gave the positive score is not championing it.

---

### Decision · Program_Chairs · 2025-01-22

Reject